# "*A custome lothsome*": Investigating the association between tobacco consumption and respiratory inflammation in two post-medieval English populations (c. CE 1500–1855)

Anna M. Davies-Barrett[1]*, Maia Casna[2], Sarah A. Inskip[1]

1 School of Archaeology and Ancient History, University of Leicester, Leicester, United Kingdom, 2 Faculty of Archaeology, Leiden University, Leiden, Netherlands

* a.daviesbarrett@leicester.ac.uk

## Abstract

Despite current clinical knowledge of the risks associated with tobacco consumption, the bioarchaeological investigation of tobacco's effect on health in past populations remains woefully underexamined. This study explores the potential respiratory health implications of the rapid incorporation of tobacco-use into the everyday lives of English citizens during the post-medieval period. Adult skeletons from urban post-medieval St James's Gardens Burial Ground, Euston, London (N = 281; CE1789–1853) and rural post-medieval (N = 151; CE1500–1855) and medieval (N = 62; CE1150–1500) Barton-upon-Humber were examined. Individuals were assessed for tobacco consumption status using osteoarchaeological and biomolecular methods. Individuals were observed for bone changes related to inflammation within the maxillary sinuses and within the pleural/pulmonary regions. Statistical tests revealed a significant association between tobacco consumption and the presence of pulmonary/pleural inflammation in the Barton-upon-Humber post-medieval group. Tobacco consumers at Barton-upon-Humber were also more than twice as likely to present with maxillary sinusitis or pleural/pulmonary inflammation, although the results were not statistically significant. Differences between tobacco consumers and non-consumers in the London group were not apparent, but the odds of having maxillary sinusitis increased by two-fold in middle adults (compared to young adults) and lower socio-economic groups (compared to higher socio-economic groups). Significant differences in respiratory disease frequencies were apparent between rural and urban groups. The results highlight the complexity of factors affecting upper and lower respiratory disease, indicating the potential impacts of not only tobacco consumption, but household, environmental, and occupational air pollution, as well as poor water sanitation, on frequencies of respiratory disease in different population groups.

**Data availability statement:** The data underlying the results presented in the study are provided to download alongside the study as Supplemental information.

**Funding:** This research was mainly funded by a UKRI-FLF research grant (grant no.: MR/T022302/1), awarded to SI. Additional funding came from a Nederlandse Organisatie voor Wetenschappelijk Onderzoek (Dutch Research Council; NWO) grant (grant no.: PGW.21.008), awarded to MC. The funders had no role in study design, data collection and analysis, decision to publish, or preparation of the manuscript.

**Competing interests:** The authors have declared that no competing interests exist.

## Introduction

"A custome lothsome to the eye, hatefull to the nose, harmefull to the braine, [and] dangerous to the lungs…" (King James I, A counterblaste to tobacco, 1604).

In 1604, King James I of England launched his *counterblaste to tobacco*, a scathing critique of the habit of smoking that had swept the country since its introduction in the mid-sixteenth century [1]. In it, King James railed against the health impacts of tobacco smoking, which he claimed affected the "*inward parts of men, soiling and infecting them, with an unctuous and oily kinde of soote, as hath bene found in some great tobacco takers, that after their death were opened*" [2]. Whilst the *counterblaste* was, in part, financially motivated by the justification of a steep import tax on tobacco [1], it is evident that the respiratory health implications of tobacco-use, in particular smoking, were apparent from the outset. Yet King James's efforts failed to curb the enthusiasm of the populace for the drug, who swiftly incorporated it into the regular facets of everyday life [3,4]. While tobacco-use occurred frequently across all strata of society, the way people consumed the intoxicant became intimately woven into the social fabric of England, usage norms being dictated by age, gender, class, occupation, and regional background [5–7]. Changes in the consumption of tobacco also varied over time according to fluctuations in fashions, politics, and the availability of tobacco products [4,8]. Although pipe smoking was common across the seventeenth to nineteenth centuries, the popularity of snuff and cigars rose and fell, and the mass-produced cigarette only broke into the market at the end of the nineteenth century [4,9]. The appetite for the commodity itself, however, never wavered, despite later efforts by Victorian anti-smoking groups to highlight the 'evils' of the habit [9].

Today, there are countless clinical studies which demonstrate that tobacco smoking produces adverse respiratory health risks. It induces pulmonary inflammation and damages the tissues of the respiratory tract, increasing susceptibility to a range of infectious and non-infectious respiratory diseases [10–12]. This includes an enhanced risk of infectious respiratory diseases from bacteria, fungi and viruses [13], including a greater likelihood of developing tuberculosis [14–16], pneumonia [17], interstitial lung disease [18], and sinusitis [19–22]. The increased risk of lung cancer, in particular, but also other cancers, such as those of the oral and digestive tract, in smokers is well established [23]. Furthermore, smoking has a systemic impact on the body, raising the risk of developing cardiovascular disease, diabetes, Alzheimer's disease, and oral diseases [24–27], to name but a few conditions associated with consumption of the drug. We are now also aware of the ways in which tobacco-related disease development can be impacted by a number of factors, such as age, sex, ethnicity, and socio-economic status, due, in part, to the interplay between disparities in access to resources brought about by one's identity and circumstances [28–30]. Cognizance of the health impacts of tobacco consumption today is well established, but eighteenth and nineteenth century warnings by medical practitioners of the risks associated with the intoxicant were largely ignored by the public. Whilst some decried

the 'immoderate' use of the drug, for example in the 'Great Tobacco Controversy' of the 1850s [9,31], for others it represented a 'panacea' to multiple ailments and was consumed in massive quantities [4].

Despite current clinical knowledge, bioarchaeological investigations of the impact of tobacco consumption on health in past populations remain woefully scarce. The identification of likely tobacco smokers in post-medieval groups is possible through the observation of pipe-notches – circular abrasions on the dentition caused by the clenching of a clay pipe between the teeth – and tobacco staining on dental lingual surfaces [5,32] (Fig 1). By identifying smokers using the presence of pipe-notches, Geber and Murphy [33] and Inskip et al. [34] have found increased frequencies of some oral diseases in pipe-smokers in Irish and Dutch populations, respectively. Walker and Henderson [32] identified increased incidences of pulmonary/pleural inflammation in tobacco-users from the cemetery of St Mary and St Michael, London. Most recently, Casna et al. [35] investigated rates of respiratory disease in tobacco-users and non-users in the Netherlands but found minimal associations between tobacco-use and respiratory disease.

Despite interesting results, a major limiting factor for research on this topic is the complex relationship between the different and often interrelated aetiologies of pathological bone changes. Greater research is required to investigate the confounding factors involved in the development of different pathological changes within the skeleton, and how these may or may not intersect with exposure to tobacco. Intersectional theory is the understanding that systems of power and inequality will result in people being differentially susceptible to the development of disease as a result of the interrelationship of aspects of their identity, such as age, gender, socio-economic status, and race, that will impact (for better or worse) their exposure to pathogens and their ability to access resources, adequate nutrition, and societal support [36]. Such an approach can aid in a better understanding of the complexity of different confounders and their relationship to tobacco. The explicit incorporation of intersectional theory in bioarchaeology is a relatively recent development, the term stemming from the work of critical race theorist Kimberlé Crenshaw [37], although certain aspects of the concept are often incorporated within bioarchaeological studies without the use of the term 'intersectional' [38,39].

A further reason for the lack of bioarchaeological research on this topic may be that traditional osteoarchaeological methods for identifying tobacco-users have proven difficult to apply to populations with high rates of tooth damage or loss, either ante- or post-mortem. In particular, rates of antemortem tooth loss in post-medieval groups appear elevated compared to previous periods [40]. However, it has now been proven possible to identify the consumption of tobacco in archaeological individuals using biomolecular methods applied to human bone [41,42]. This approach runs small samples (<50 mg) of human femoral cortical bone through a metabolomic workflow, producing information on the antemortem and postmortem metabolomic profile of the individual. It has now been shown that tobacco consumption status can be assigned to individuals with around 90% accuracy. This is achieved through the statistical comparison of metabolomic profiles from individuals with unknown consumption status to those from groups of osteoarchaeologically identified tobacco consumers, those with no osteoarchaeological evidence for tobacco consumption, and control groups pre-dating the use of tobacco [42], revolutionising our understanding of who was, and was not, smoking in the past.

Utilising this new technique, we can now ask with greater confidence: what might have been the health implications of the rapid incorporation of tobacco-use into the everyday lives of English citizens? Might we see a disparity in respiratory disease rates in different social groups according to smoking status and factors such as age, sex, social status, and regionality, resulting from intersectional aspects of identity [38]? To explore these questions, we formulated two hypotheses based on our clinical and sociological understandings of the impact of tobacco on respiratory health:

1. Tobacco consumers will demonstrate a higher prevalence of respiratory disease than non-consumers.

2. Tobacco consumption status will intersect with aspects of identity–age, sex, socio-economic status, and regionality–to influence respiratory disease profiles in different groups.

We tested these hypotheses by examining the frequency of upper and lower respiratory tract disease in identified tobacco-users and non-users from two disparate but contemporaneous English post-medieval (c. CE 1500–1855)

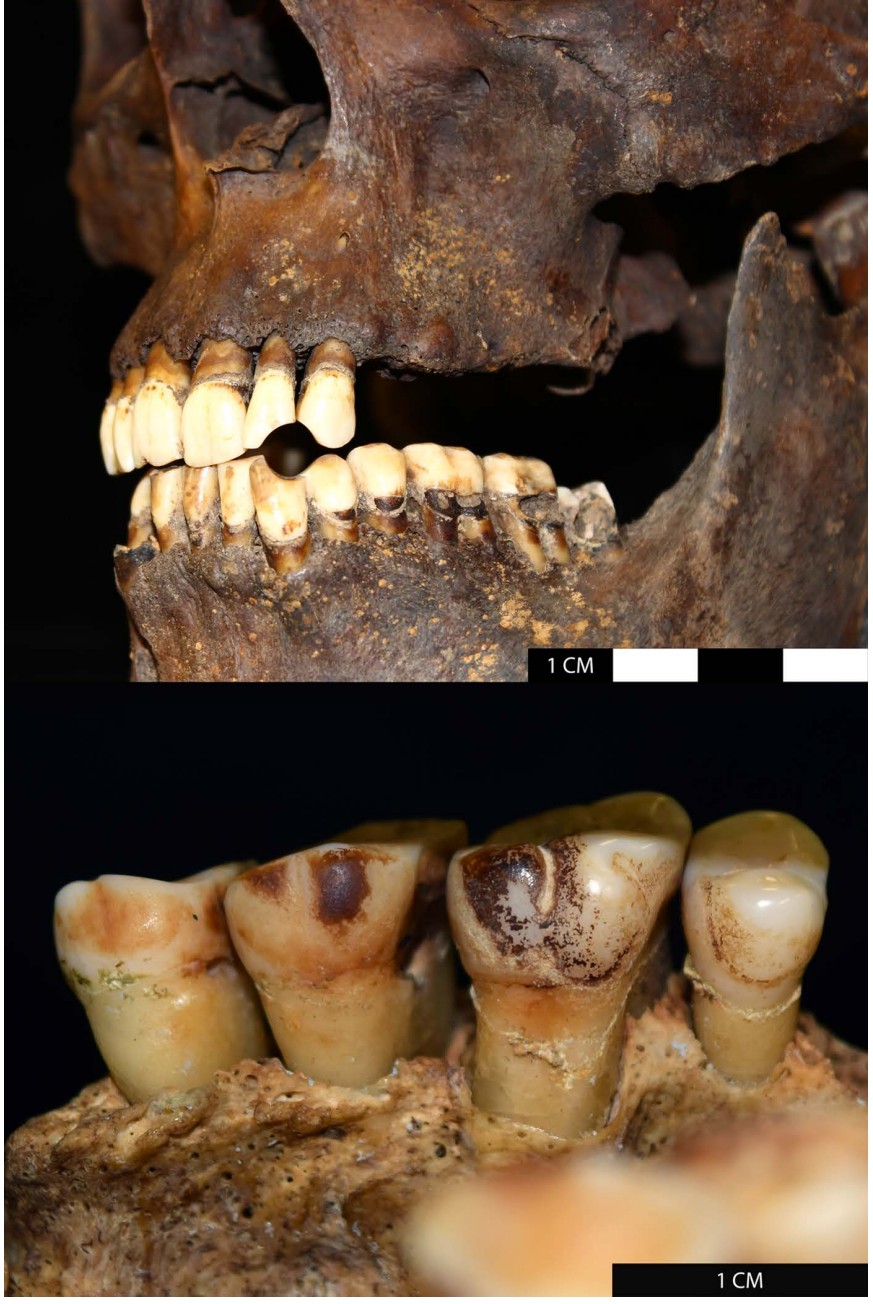

**Fig 1. Osteoarchaeological evidence for tobacco consumption.** Top: A 'pipe-notch' – a circular abrasion to the teeth caused by the habitual clenching of a clay pipe. Bottom: tobacco staining of the lingual (inner) surfaces of the teeth (adapted from Figs 6.1 and 6.2 in [5], available under a CC-BY 4.0 license).

populations: skeletal individuals from Barton-upon-Humber, a rural, mainly agricultural group, and from St James's Gardens Burial Ground, Euston, an urban group made up of people from different social strata in London. The results are statistically analysed to investigate the potential impacts of tobacco on respiratory health among the varying social groups of post-medieval England.

## Populations

The town of Barton-upon-Humber is located in North Lincolnshire, opposite the city of Hull, on the banks of the river Humber (Fig 2). During the post-medieval period, it appears that the residents of Barton-upon-Humber were, until the late eighteenth century, mainly land owners and agricultural tenants [43,44]. Upheavals in the farming system and the introduction of the railway to the town led to mass expansion, with the population doubling to 3,866 people between 1801 and 1850 [45]. A variety of male employments existed during the nineteenth century, consisting mainly of farmers, labourers, brick and tile makers, shopkeepers, and traders, with women limited to domestic or craft services [43]. St Peter's Church served the local populace as a cemetery for almost a millennium, containing burials spanning the late tenth century until CE 1855 [46]. The cemetery was excavated between 1978 and 1984 and skeletal individuals were, at the time of analysis, housed within St Peter's Church, at Barton-upon-Humber. From the cemetery, a sample of 132 post-medieval (CE

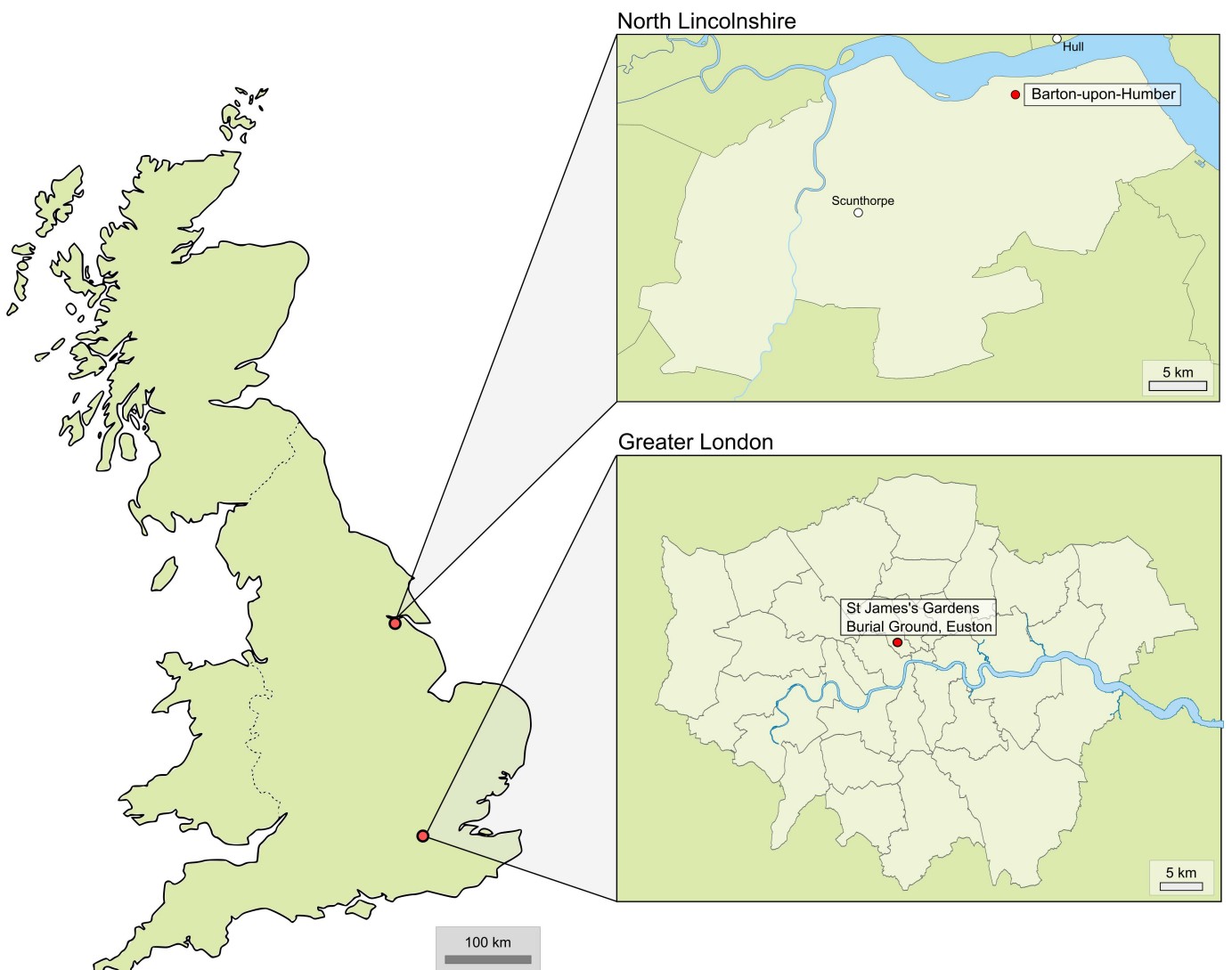

**Fig 2. Map of Great Britain with the location of populations within the current study (from Fig 1 in [47], available under a CC-BY 4.0 license).**

1500–1855) adults were analysed, as well as a smaller comparative medieval (CE 1150–1500) sample of 62 adults, to provide a baseline for respiratory disease frequencies prior to the introduction of tobacco.

St James's Gardens Burial Ground, located in Euston, London, was in use between CE 1789 and 1853, serving the local parish of St James's Church, Piccadilly, as well as local non-parishioners [48]. During this period, the socio-economic status (SES) of its inhabitants was mixed [49, Map Descriptive of London Poverty 1898–1899. Sheet 6.] and the parish cemetery contained a range of occupants, from the echelons of high society to paupers' graves [48]. Westminster poll books for male rate payers (those who owned property), dating from between CE 1790 and 1820, demonstrate the huge range of occupations present among men in the St James's parish, including among many other roles: 'gentlemen'; surgeons and physicians; booksellers; jewellers and silversmiths; chandlers; poulterers, butchers, cheesemongers, bakers, grocers, victuallers and merchants of wine, oysters, etc.; carpenters and cabinet makers; stable keepers; and haberdashers, hatters, tailors, and drapers, etc. [50]. As these records belong to male 'rate payers', these occupations tend to represent the upper to middling sorts, yet they demonstrate the great variety in economic specialisations found within the area at the time [51]. The area was not without its poor; it was reported in c. 1730 that there were at least 1000 within the parish [52] and the lowest status burial ground was dedicated to paupers' graves (those who could not afford their own funeral). Excavated between 2017 and 2021, a sample of 281 adult individuals were included in the current study from St James's Gardens Burial Ground. At the time of analysis, these individuals were under the guardianship of Mace Dragados in partnership with HS2 and were housed within storage in Heathrow, Middlesex. The SES of these individuals can be inferred from the geographical location of their burials within the grounds. Four burial grounds are apparent from burial records, the differing fees for which are likely to have produced some socio-economic division in the populations buried within them. These burial grounds are likely to consist in the majority of high (burial ground 1, closest to the chapel), upper-middle (burial ground 2), lower-middle (burial ground 3) and lower (burial ground 4, the pauper's cemetery) class groups [48]. However, the allocation of status according to burial ground may not account for some social mobility during life or in the choice of burial ground by relatives after death, particularly among the working and 'middling' classes.

No permits were required for the described study, which complied with all relevant regulations. All individuals included in the current study were temporarily transported to the Human Osteology Laboratory at the University of Leicester for analysis.

## Methods

Only individuals with a minimum of the presence of one maxillary sinus or at least 25% of the ribcage present were included within the final analysis. From the Barton-upon-Humber sample, 132 of the 153 post-medieval individuals (86.3%) and 60 of the 62 medieval individuals (96.8%) originally assessed were able to be included in the final analysis. From the St James's Garden Burial Ground sample, 254 of the 281 individuals (90.4%) originally assessed were able to be included in the final analysis. All relevant raw data pertaining to the individuals included in the final analysis can be accessed in S1 File.

All individuals were assessed for evidence of tobacco consumption using a combination of osteoarchaeological and biomolecular methods. Osteoarchaeological identification of tobacco consumers consisted of the identification of pipe-notches and/or lingual staining on the dentition according to the methods presented by Davies-Barrett and Inskip [5]. It should be noted that an absence of evidence for pipe-notches or lingual staining does not necessarily indicate that an individual was completely unexposed to tobacco. Individuals with no pipe-notches or staining, who we refer to in the current paper as 'non-consumers', may have consumed tobacco infrequently, resulting in a lack of dental evidence for tobacco consumption, or may have been exposed to tobacco via passive inhalation of smoke. We assume, therefore, that those with dental evidence for tobacco consumption were habitual users, whereas those without evidence were infrequently or not at all exposed. The intensity of use of tobacco smoking has, today, been linked to increased risk of developing respiratory disease [53]. Further, there has been shown to be a greater risk of developing respiratory disease in active smokers

than passive smokers (although passive smokers are still at greater risk than unexposed individuals) [17]. Therefore, if our hypotheses are confirmed, we might expect a higher frequency of respiratory inflammation in habitual tobacco consumers.

For individuals for whom it was not possible to identify tobacco consumption status osteoarchaeologically (due to the absence of dentition), biomolecular methods were applied. This relied on the metabolomic identification of small molecules (metabolites) in femoral bone samples, which were statistically compared to the molecular signatures from identified tobacco consumers and non-consumers from the same populations, allowing tobacco consumption status to be estimated. The laboratory protocols and results of this analysis are supplied in Badillo-Sanchez et al. [42], which made use of the same individuals as that included in the current study. All biomolecular tobacco assignations used in the current study were taken from the data provided in Supplementary File S4 of Badillo-Sanchez et al. [42]. Due to instances of poor preservation, it was not possible to determine smoking status for every individual in the sample. It was possible to allocate tobacco consumption status to 198 of the 386 post-medieval individuals (51.3%) included in the final analysis using osteo-archaeological methods, and a further 115 individuals (29.8%) using biomolecular methods. However, 73 post-medieval individuals (18.9%) analysed for respiratory disease could not be allocated to a consumption status group due to poor preservation of both the dentition and the femur or due to a failure during the biomolecular analysis. Pathology results from individuals with unidentified consumption status are included in the total group category.

Age was estimated using standard osteoarchaeological methods involving observation of the pubis, auricular surface, sternal rib ends, and sternal clavicle joint surface [54–59]. Only adults were analysed as part of the current study due to the unknown effects of younger ages on metabolite signatures. Individuals were assigned into one of four age group categories, after Buikstra and Ubelaker [60]: young adult (20–34 years), middle adult (35–49 years), mature adult (50 + years), or unknown adult (20 + years). Skeletal sex was assessed by observing the morphology of the skull and pelvis [60–64]. Individuals were assigned as either probable female, possible female, probable male, possible male, indeterminate, or unobservable for sex. These groups were later pooled to facilitate statistical analysis into the following groups: female (including all probable/possible females), male (including all probable/possible males), and unknown (including all unobservable/indeterminate individuals). To allow meaningful statistical analysis, high and high-middle SES groups were combined into a high-middle group, and low and low-middle SES groups into a low-middle group.

A discussion of the pathophysiological processes leading to new bone on the visceral surfaces of the ribs and changes within the maxillary sinuses can be found in greater detail in Davies-Barrett et al. [65,66]. Periosteal new bone formation on the ribs (thought to represent pulmonary/pleural inflammation) (Fig 3) was assessed according to Davies-Barrett et al. [65]. All individuals with 25% or more of the rib cage present were observed for new bone on the visceral surfaces of the ribs. Bone changes within the maxillary sinuses (thought to represent chronic sinusitis) were assessed according to Davies-Barrett et al. [66], based on methods by Boocock et al. [67]. All individuals with the presence of at least one sinus, 25% or more complete, were observed for bone changes within the maxillary sinus indicative of chronic inflammation. Individuals were recorded as present for maxillary or pulmonary/pleural inflammation if any of the relevant bone changes presented in Davies-Barrett et al. [65,66] were observed within the sinuses or on the visceral surfaces of the ribs. A Karl Storz SE & Co. KG digital tip flexible endoscope and Tele-pak X were used to visualise the internal surfaces of intact maxillary sinuses. This equipment was applied only in instances where taphonomic damage allowed access of the endoscope to the sinus.

Statistical analysis was undertaken using SPSS Statistics V.28. A chi-square test [$X^2$] was applied to investigate differences in the prevalence of disease between groups. This test included the production of adjusted standardised residuals [AR], which measure the strength of difference between observed values and expected values. A standardised residual value of -/+ 2 represents a frequency that is lower/higher than is expected by two standard deviations, indicating a major contributor towards the difference observed between groups. As multiple chi-square tests were undertaken on the same datasets, it was decided that a correction factor should be applied. The traditional Bonferroni correction, which divides the significance level by the number of tests performed, has been critiqued for being unnecessarily conservative [68].

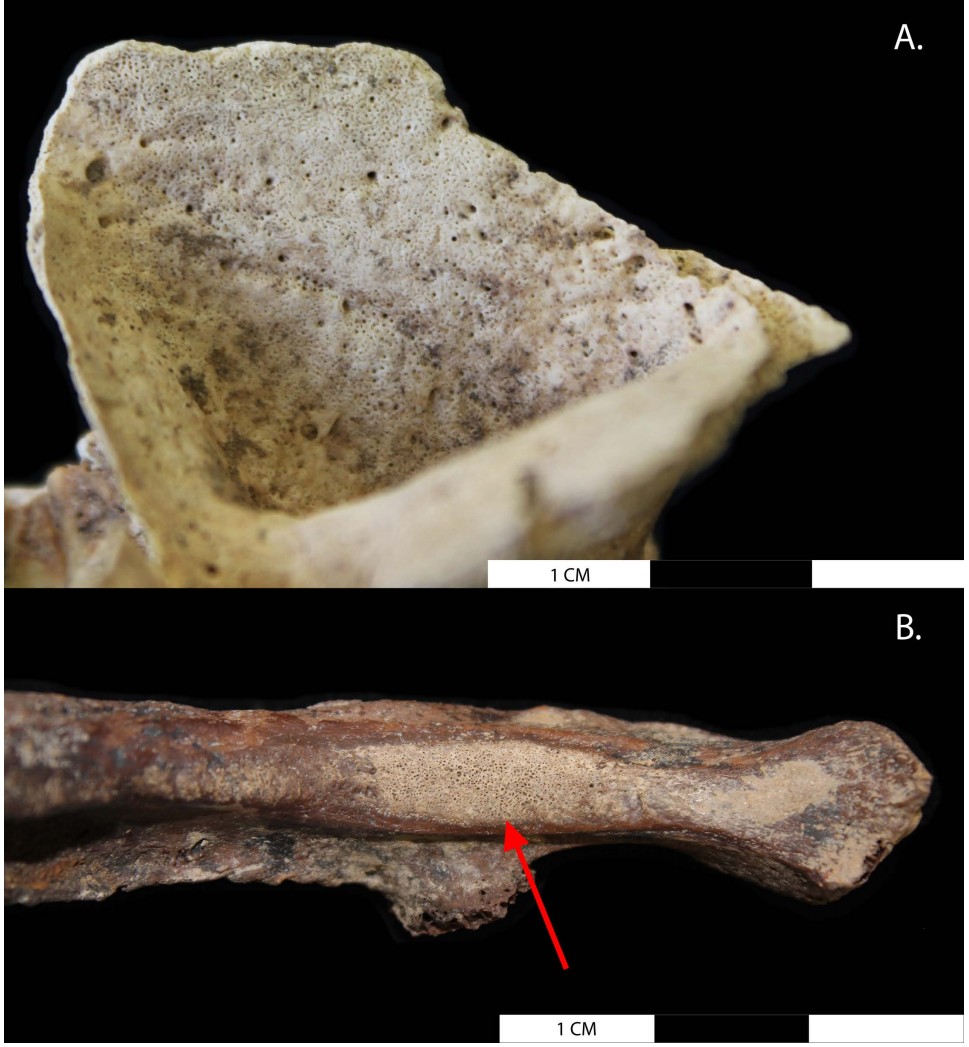

**Fig 3. Evidence for respiratory inflammation in human skeletons. A.** Porous new bone formation on the walls of the right maxillary sinus in a mature adult male from Barton-upon-Humber (Sk911). **B.** Woven bone on the neck of the right fourth rib in a middle adult female from St James's Gardens Burial Ground (Sk112237).

An adaptation of this correction, known as the Holm-Bonferroni method, is considered more powerful [69] and was, thus, applied. This approach compares each *p*-value to a critical threshold by ranking the *p*-values based on the order of significance and producing an adjusted *p*-value in relation to its rank among other *p*-values and the number of tests undertaken [69,70].

Binary logistic regression was applied to produce the odds ratio (OR) of the presence of respiratory disease in different tobacco consumption status, age, skeletal sex, and SES groups. In this instance, the OR indicates the likelihood of one group being less or more likely to be associated with the presence of a disease. In binary logistic regression, if the upper limit of the 95% confidence interval falls below an OR value of 1, then a group is considered to be significantly less likely to be associated with the disease than the other group. If the lower limit of the 95% confidence interval falls above an OR value of 1, then a group is considered to be significantly more likely to be associated with the disease than the other group. In all statistical analysis, a *p*-value of ≤0.05 was considered to be statistically significant.

All research undertaken as part of the current study was approved by the University of Leicester Ethics Committee. All treatment of human remains was undertaken according to the Codes of Ethics and Practice provided by the British Association for Biological Anthropology and Osteoarchaeology.

## Results

Demographic data for each archaeological group can be found in Fig 4 (S1 File provides raw data). Almost all age, skeletal sex, tobacco consumption, and SES groups were well represented. However, a lower number of mature adults was present in the medieval Barton-upon-Humber group. This was also the case for mature adults and individuals of high socio-economic status in the St James's sample group compared to other age and SES groups in this population. This may have had some impact on statistical analysis pertaining to these groups.

While the prevalence rate of maxillary sinusitis was 15.4% higher within the post-medieval Barton-upon-Humber group compared to the St James's population, the opposite was observed in the prevalence rate of pulmonary/pleural inflammation, which was 19.3% higher in the St James's group (Tables 1 and 2). Both results were statistically significant prior to application of a Holm-Bonferroni correction (Table 3); however, only a difference in pulmonary/pleural inflammation between sites remained below the significance threshold after correction for multiple tests ($X^2(1)=11.623$, $p=<.001$; adjusted $p=.004$). Prevalence rates for both pulmonary/pleural inflammation and maxillary sinusitis were higher in tobacco consumers within the post-medieval Barton-upon-Humber population. Further, the medieval sample group presented with

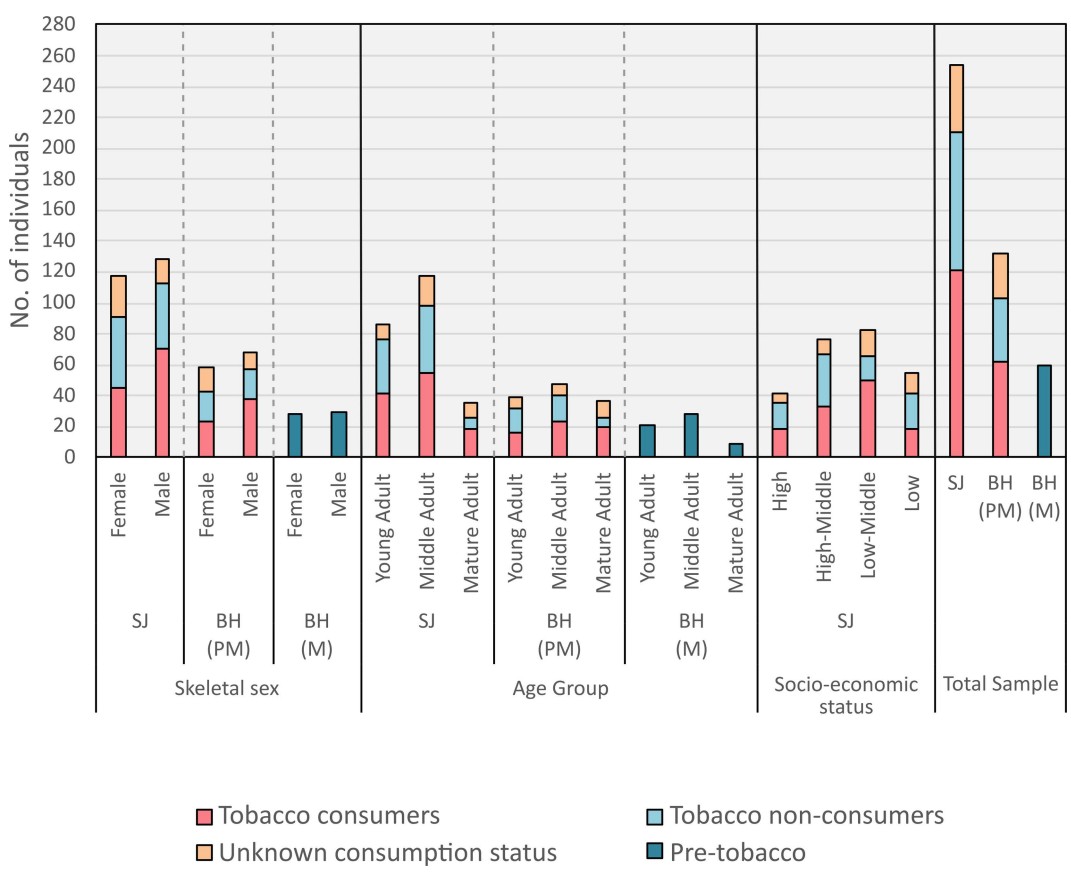

**Fig 4. Demographic make-up of the sample.** SJ = St James's Gardens Burial Ground; BH(PM) = Post-medieval Barton-upon-Humber; BH(M) = Medieval Barton-upon-Humber.

Maxillary sinusitis

**Table 1. Prevalence rates of maxillary sinusitis in different archaeological groups according to tobacco consumption status, skeletal sex, estimated age, and socio-economic status.**

| Sex | Age | Barton-upon-Humber | | | | St James's Gardens Burial Ground | | | | | | | | |
| --- | --- | --- | --- | --- | --- | --- | --- | --- | --- | --- | --- | --- | --- | --- |
| | | | | | | All | | | High-middle socioeconomic status | | | Low-middle socioeconomic status | | |
| | | Tobacco non-consumers | Tobacco consumers | All post-medieval sample group | All medieval sample group | Tobacco non-consumers | Tobacco consumers | All | Tobacco non-consumers | Tobacco consumers | All | Tobacco non-consumers | Tobacco consumers | All |
| **Female** | Young adult | 71.4% 5/7 | 66.7% 2/3 | 69.2% 9/13 | 50.0% 4/8 | 31.6% 6/19 | 58.3% 7/12 | 41.7% 15/36 | 18.2% 2/11 | 50.0% 2/4 | 26.7% 4/15 | 50.0% 4/8 | 62.5% 5/8 | 52.4% 11/21 |
| | Middle adult | 50.0% 2/4 | 83.3% 5/6 | 69.2% 9/13 | 50.0% 5/10 | 63.2% 12/19 | 72.7% 8/11 | 63.2% 24/38 | 53.8% 7/13 | 66.7% 2/3 | 55.0% 11/20 | 83.3% 5/6 | 75.0% 6/8 | 72.2% 13/18 |
| | Mature adult | 100% 2/2 | 100% 4/4 | 85.7% 6/7 | 100% 1/1 | 100% 1/1 | 20.0% 2/10 | 43.8% 7/16 | – | 33.3% 2/6 | 44.4% 4/9 | 100% 1/1 | 0% 0/4 | 42.9% 3/7 |
| | Unknown adult | 0% 0/1 | 100% 2/2 | 50.0% 2/4 | 100% 1/1 | 50.0% 2/4 | 100% 1/1 | 57.1% 4/7 | 50.0% 2/4 | 100% 1/1 | 66.7% 4/6 | – 0/0 | – 0/0 | 0% 0/1 |
| | Total | 64.3% 9/14 | 86.7% 13/15 | 70.3% 26/37 | 55.0% 11/20 | 48.8% 21/43 | 52.9% 18/34 | 51.5% 50/97 | 39.3% 11/28 | 50.0% 7/14 | 46.0% 23/50 | 66.7% 10/15 | 55.0% 11/20 | 57.4% 27/47 |
| **Male** | Young adult | 57.1% 4/7 | 100% 4/4 | 72.7% 8/11 | 60.0% 3/5 | 35.7% 5/14 | 47.1% 8/17 | 43.8% 14/32 | 42.9% 3/7 | 50.0% 4/8 | 46.7% 7/15 | 28.6% 2/7 | 44.4% 4/9 | 41.2% 7/17 |
| | Middle adult | 100% 3/3 | 57.1% 4/7 | 75.0% 9/12 | 66.7% 4/6 | 66.7% 12/18 | 47.1% 16/34 | 50.8% 31/61 | 70.0% 7/10 | 28.6% 4/14 | 37.9% 11/29 | 62.5% 5/8 | 60.0% 12/20 | 62.5% 20/32 |
| | Mature adult | 33.3% 1/3 | 72.7% 8/11 | 64.7% 11/17 | 80.0% 4/5 | 100% 5/5 | 66.7% 6/9 | 81.3% 13/16 | 100% 1/1 | 25.0% 1/3 | 50.0% 3/6 | 100% 4/4 | 100% 5/5 | 100% 10/10 |
| | Unknown adult | 100% 1/1 | 0% 0/1 | 50.0% 2/4 | – 0/0 | – 0/0 | 100% 2/2 | 100% 4/4 | – 0/0 | – | – 0/0 | – | 100% 2/2 | 100% 4/4 |
| | Total | 64.3% 9/14 | 69.6% 16/23 | 68.2% 30/44 | 68.8% 11/16 | 59.5% 22/37 | 51.6% 32/62 | 54.9% 62/113 | 61.1% 11/18 | 34.6% 9/26 | 42.0% 21/50 | 57.9% 11/19 | 63.9% 23/36 | 65.1% 41/63 |
| **Total** | Young adult | 66.7% 10/15 | 75.0% 6/8 | 70.4% 19/27 | 50.0% 7/14 | 33.3% 11/33 | 50.0% 15/30 | 42.0% 29/69 | 27.8% 5/18 | 46.2% 6/13 | 35.5% 11/31 | 40.0% 6/15 | 52.9% 9/17 | 47.4% 18/38 |
| | Middle adult | 71.4% 5/7 | 69.2% 9/13 | 69.2% 18/26 | 52.9% 9/17 | 64.9% 24/37 | 53.2% 25/47 | 55.4% 56/101 | 60.9% 14/23 | 38.9% 7/18 | 46.0% 23/50 | 71.4% 10/14 | 62.1% 18/29 | 64.7% 33/51 |
| | Mature adult | 60.0% 3/5 | 80.0% 12/15 | 70.8% 17/24 | 83.3% 5/6 | 100% 6/6 | 42.1% 8/19 | 62.5% 20/32 | 100% 1/1 | 30.0% 3/10 | 46.7% 7/15 | 100% 5/5 | 55.6% 5/9 | 76.5% 13/17 |
| | Unknown adult | 50.0% 1/2 | 66.7% 2/3 | 50.0% 4/8 | 100% 1/1 | 60.0% 3/5 | 60.0% 3/5 | 64.3% 9/14 | 50.0% 2/4 | 100% 1/1 | 66.7% 4/6 | 100% 1/1 | 50% 2/4 | 62.5% 5/8 |
| | Total | 65.5% 19/29 | 74.4% 29/39 | 68.2% 58/85 | 57.9% 22/38 | 54.3% 44/81 | 50.5% 51/101 | 52.8% 114/216 | 47.8% 22/46 | 40.5% 17/42 | 44.1% 45/102 | 62.9% 22/35 | 57.6% 34/59 | 60.5% 69/114 |

Table 2. Prevalence rates of pulmonary/pleural inflammation in different archaeological groups according to tobacco consumption status, skeletal sex, estimated age, and socio-economic status.

Pulmonary/pleural inflammation

| Sex | Age | Barton-upon-Humber | | | | St James's Gardens Burial Ground — All | | | High-middle socioeconomic status | | | Low-middle socioeconomic status | | |
|---|---|---|---|---|---|---|---|---|---|---|---|---|---|---|
| | | Tobacco non-consumers | Tobacco consumers | All post-medieval sample group | All medieval sample group | Tobacco non-consumers | Tobacco consumers | All | Tobacco non-consumers | Tobacco consumers | All | Tobacco non-consumers | Tobacco consumers | All |
| **Female** | Young adult | 25.0% 2/8 | 28.6% 2/7 | 25.0% 4/16 | 10.0% 1/10 | 52.9% 9/17 | 31.6% 6/19 | 46.5% 20/43 | 40.0% 4/10 | 25.0% 2/8 | 33.3% 6/18 | 71.4% 5/7 | 36.4% 4/11 | 56.0% 14/25 |
| | Middle adult | 11.1% 1/9 | 28.6% 2/7 | 15.8% 3/19 | 15.4% 2/13 | 50.0% 11/22 | 40.0% 6/15 | 50.0% 23/46 | 43.8% 7/16 | 50.0% 2/4 | 50.0% 12/24 | 66.7% 4/6 | 36.4% 4/11 | 50.0% 11/22 |
| | Mature adult | 0% 0/1 | 33.3% 1/3 | 18.2% 2/11 | 0% 0/2 | 0% 0/1 | 60.0% 6/10 | 42.1% 8/19 | – 0/0 | 50.0% 3/6 | 40.0% 4/10 | 0% 0/1 | 75.0% 3/4 | 44.4% 4/9 |
| | Unknown adult | – 0/0 | 0% 0/2 | 0% 0/2 | 0% 0/1 | – 0/0 | – 0/0 | 0% 0/2 | – 0/0 | 0% 0/1 | 0% 0/1 | – 0/0 | – 0/0 | 0% 0/1 |
| | Total | 16.7% 3/18 | 26.3% 5/19 | 18.8% 9/48 | 11.5% 3/26 | 50.0% 20/40 | 40.9% 18/44 | 46.4% 51/110 | 42.3% 11/26 | 36.8% 7/19 | 41.5% 22/53 | 64.3% 9/14 | 42.3% 11/26 | 50.9% 29/57 |
| **Male** | Young adult | 0% 0/6 | 75% 6/8 | 43.8% 7/16 | 12.5% 1/8 | 53.3% 8/15 | 52.4% 11/21 | 54.1% 20/37 | 75.0% 6/8 | 54.5% 6/11 | 63.2% 12/19 | 28.6% 2/7 | 50.0% 5/10 | 44.4% 8/18 |
| | Middle adult | 28.6% 2/7 | 30.8% 4/13 | 27.3% 6/22 | 9.1% 1/11 | 57.9% 11/19 | 36.4% 12/33 | 41.9% 26/62 | 66.7% 6/9 | 33.3% 5/15 | 42.9% 12/28 | 50.0% 5/10 | 38.9% 7/18 | 41.2% 14/34 |
| | Mature adult | 50.0% 1/2 | 33.3% 4/12 | 35.3% 6/17 | 0% 0/7 | 20.0% 1/5 | 33.3% 3/9 | 37.5% 6/16 | 0% 0/1 | 50.0% 2/4 | 50.0% 3/6 | 25.0% 1/4 | 20.0% 1/5 | 30.0% 3/10 |
| | Unknown adult | – 0/0 | 0% 0/1 | 0% 0/2 | 0% 0/1 | – 0/0 | – 0/0 | 50% 1/2 | – 0/0 | – 0/0 | – 0/0 | – 0/0 | – 0/0 | 50.0% 1/2 |
| | Total | 20.0% 3/15 | 41.2% 14/34 | 33.3% 19/57 | 7.4% 2/27 | 51.3% 20/39 | 41.3% 26/63 | 45.3% 53/117 | 66.7% 12/18 | 43.3% 13/30 | 50.9% 27/53 | 38.1% 8/21 | 39.4% 13/33 | 40.6% 26/64 |
| **Total** | Young adult | 13.3% 2/15 | 50.0% 8/16 | 36.1% 13/36 | 10.0% 2/20 | 53.1% 17/32 | 41.5% 17/41 | 50.6% 42/83 | 55.6% 10/18 | 40.0% 8/20 | 47.4% 18/38 | 50.0% 7/14 | 42.9% 9/21 | 53.3% 24/45 |
| | Middle adult | 17.6% 3/17 | 30.0% 6/20 | 20.9% 9/43 | 12.0% 3/25 | 53.7% 22/41 | 38.8% 19/49 | 45.9% 50/109 | 52.0% 13/25 | 40.0% 8/20 | 47.2% 25/53 | 56.3% 9/16 | 37.9% 11/29 | 44.6% 25/56 |
| | Mature adult | 33.3% 1/3 | 33.3% 5/15 | 28.6% 8/28 | 0% 0/9 | 16.7% 1/6 | 47.4% 9/19 | 40.0% 14/35 | 0% 0/1 | 50.0% 5/10 | 43.8% 7/16 | 20.0% 1/5 | 44.4% 4/9 | 36.8% 7/19 |
| | Unknown adult | – 0/0 | 0% 0/3 | 0% 0/4 | 0% 0/2 | – 0/0 | 0% 0/1 | 25% 1/4 | – 0/0 | 0% 0/1 | 0% 0/1 | – 0/0 | – 0/0 | 33.3% 1/3 |
| | Total | 17.1% 6/35 | 35.2% 19/54 | 27.0% 30/111 | 8.9% 5/56 | 50.6% 40/79 | 40.9% 45/110 | 46.3% 107/231 | 52.3% 23/44 | 41.2% 21/51 | 46.3% 50/108 | 48.6% 17/35 | 40.7% 24/59 | 46.3% 57/123 |

**Table 3. Chi-square analysis of significant differences in maxillary sinusitis and pulmonary inflammation among different populations and consumer groups.**

| | | Maxillary sinusitis | | | | | Pulmonary inflammation | | | | |
|---|---|---|---|---|---|---|---|---|---|---|---|
| | | N | Absent n (%) [AR] | Present n (%) [AR] | Chi-square outcome | Adjusted p-value | N | Absent n (%) [AR] | Present n (%) [AR] | Chi-square outcome | Adjusted p-value |
| **Tobacco consumption status: Barton-upon-Humber** | **Medieval** | 106 | 16 (15.1) [1.3] | 22 (20.8) [-1.3] | $X^2(2)=2.331$, $p=.312$ | .624 | 145 | 51 (35.2) [2.8] | 5 (3.4) [-2.8] | $X^2(2)=11.904$, $p=.003*$ | .015* |
| | **Tobacco non-consumers** | | 10 (9.4) [.1] | 19 (17.9) [-.1] | | | | 29 (20.0) [.6] | 6 (4.1) [.6] | | |
| | **Tobacco consumers** | | 10 (9.4) [-1.4] | 29 (27.4) [1.4] | | | | 35 (24.1) [-3.3] | 19 (13.1) [3.3] | | |
| **Tobacco consumption status: St James's Gardens Burial Ground** | **Tobacco non-consumers** | 182 | 37 (20.3) [-.5] | 44 (24.2) [.5] | $X^2(1)=.264$, $p=.656$ | .656 | 189 | 39 (20.6) [-1.3] | 40 (21.2) [1.3] | $X^2(1)=1.757$, $p=.185$ | .555 |
| | **Tobacco consumers** | | 50 (27.5) [.5] | 51 (28.0) [-.5] | | | | 65 (34.4) [1.3] | 45 (23.8) [-1.3] | | |
| **Post-medieval population** | **Barton-upon-Humber (post-medieval)** | 301 | 27 (9.0) [-2.4] | 58 (19.3) [-2.4] | $X^2(1)=5.951$, $p=.015$ | .060 | 342 | 81 (23.7) [3.4] | 30 (8.8) [-3.4] | $X^2(1)=11.623$, $p=<.001*$ | .004* |
| | **St James's Gardens Burial Ground** | | 102 (33.9) [2.4] | 114 (37.9) [-2.4] | | | | 124 (36.3) [-3.4] | 107 (31.3) [3.4] | | |

AR = Adjusted residuals. *= p-value ≤.05.

the lowest prevalence rates within this population. While the difference between tobacco consumers, non-consumers, and the medieval group was not statistically significant for maxillary sinusitis ($X^2(2)=2.331$, $p=.312$; adjusted $p=.624$), a significant difference was observed for pulmonary/pleural inflammation ($X^2(2)=11.904$, $p=.003$; adjusted $p=.015$). Adjusted residuals indicate that the groups contributing to the greatest difference were the medieval and tobacco consumer groups. The prevalence of maxillary sinusitis among tobacco consumers and non-consumers within the St James's population were similar, while tobacco non-consumers presented with an 9.7% higher prevalence rate of pulmonary/pleural inflammation than consumers. No significant differences were observed in chi-square outcomes among groups from St James's.

The effect of the interaction between different variables on the likelihood of having respiratory inflammation can be investigated using hierarchical loglinear analysis. This approach allows for an intersectional analysis of the ways in which different aspects of a person's identity, as perceived through archaeological context and skeletal data, might affect the likelihood of a person developing respiratory inflammation [38]. Unfortunately, in the current study, once groups were broken down by tobacco consumption status, age group, skeletal sex, and SES, group sizes were too small for hierarchical loglinear analysis to be effective. However, prevalence from an intersectional perspective is presented in Fig 5. Although it should be borne in mind that the rates presented here are calculated from small sample sizes, some interesting patterns can be observed that indicate potential for future research.

Both male and female tobacco consumers from Barton-upon-Humber presented with higher rates of respiratory inflammation compared to non-consumers. However, female smokers, particularly in older age categories, seemed to be disproportionately affected by maxillary sinusitis, while male smokers presented with the highest rates of pulmonary/

 

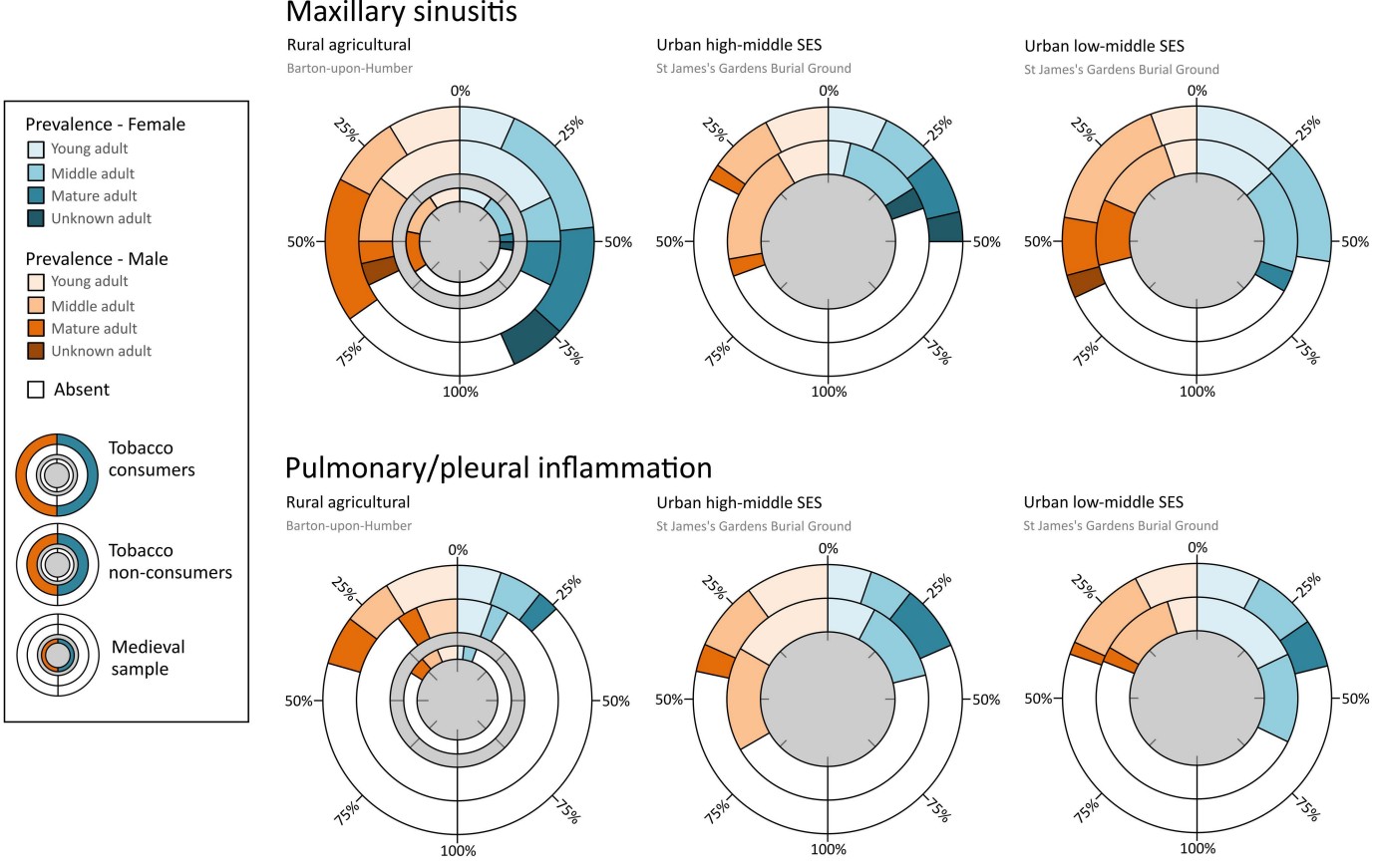

**Fig 5. Prevalence rates (%) of maxillary sinusitis and pulmonary/pleural inflammation among tobacco consumers and non-consumers in different social groups during the post-medieval period.** SES = socio-economic status.

pleural inflammation. The group from St James's that appeared least likely to develop maxillary sinusitis was high-middle SES women. Low-middle SES groups from both sexes and tobacco consumption statuses had higher rates of maxillary sinusitis, but, conversely, rates in high-middle SES non-consuming men were also comparable. Finally, those at greatest risk for pulmonary/pleural inflammation appeared to be men of high-middle SES and women of low-middle SES in young and middle age groups who did not consume tobacco.

The impact of different variables (tobacco consumption status, skeletal sex, and estimated age) on respiratory inflammation were further investigated using binary logistic regression analysis (Table 4). Results from the Barton-upon-Humber population indicate that tobacco consumers were 2.0 and 2.7 times more likely than non-consumers to present with evidence for maxillary sinusitis and pulmonary/pleural inflammation, respectively. However, due to broad confidence intervals, neither result was statistically significant and, thus, this outcome should be interpreted cautiously. Males at Barton-upon-Humber were 1.7 times more likely to present with evidence of pulmonary/pleural inflammation than females; again, this was not statistically significant. No statistically significant increase in the odds ratio of respiratory disease was found among any of the age groups at Barton-upon-Humber. Within the St James's population, there was no increased likelihood of pulmonary/pleural inflammation in any category. However, a two-fold increase in the likelihood of having maxillary sinusitis was present in middle and older age categories, when compared to the young adult category. This result was significant ($p = .042$) in the middle adult age category. Further, there was a two-times greater likelihood of having maxillary sinusitis in the lower-middle SES group, when compared to the high-middle SES group ($p = .020$).

**Table 4. Outcomes of binary logistic regression.**

| | | Maxillary sinusitis | | | | Pulmonary inflammation | | | |
| --- | --- | --- | --- | --- | --- | --- | --- | --- | --- |
| | | OR | CI for OR | | p-value | OR | CI for OR | | p-value |
| | | | Lower | Upper | | | Lower | Upper | |
| Barton-upon-Humber (post-medieval only) | Tobacco consumption status | 2.003 | .583 | 6.884 | .270 | 2.706 | .898 | 8.156 | .077 |
| | Skeletal sex | .575 | .173 | 1.913 | .367 | 1.696 | .597 | 4.823 | .322 |
| | Age | | | | .890 | | | | .577 |
| | Middle adult | .736 | .176 | 3.071 | .674 | .565 | .184 | 1.733 | .318 |
| | Mature adult | .986 | .219 | 4.449 | .986 | .602 | .157 | 2.305 | .459 |
| St James's Gardens Burial Ground | Tobacco consumption status | .711 | .369 | 1.370 | .309 | .694 | .382 | 1.262 | .231 |
| | Skeletal sex | 1.084 | .571 | 2.061 | .805 | 1.043 | .579 | 1.879 | .889 |
| | Age | | | | .112 | | | | .891 |
| | Middle adult | 2.042 | 1.025 | 4.067 | .042* | .904 | .483 | 1.695 | .754 |
| | Mature adult | 1.920 | .720 | 5.121 | .192 | .805 | .316 | 2.051 | .649 |
| | Socio-economic status | 2.111 | 1.122 | 3.970 | .020* | .919 | .512 | 1.650 | .778 |

Constants: smoking status = non-tobacco consumers; skeletal sex = females; age = young adult; socio-economic status = high-middle. All individuals with unknown age, skeletal sex, tobacco consumption, and respiratory disease status have been removed from sample. Acronyms: OR = Odds ratio; CI = Confidence intervals. * = p-value ≤ .05.

## Discussion

Hypothesis 1, that tobacco consumers would demonstrate higher frequencies of respiratory inflammation than their non-smoking contemporaries, was only proved for pulmonary/pleural inflammation at Barton-upon-Humber. Given the strong clinical case for the adverse effects of tobacco-use on respiratory health, the lack of statistically significant differences in the current study between tobacco consumers and non-consumers may seem surprising. Further, it remained difficult to prove our second hypothesis–that different aspects of identity would work in conjunction with tobacco consumption status to dictate the respiratory disease profile of different groups–due to the small sample sizes produced by breaking down groups using multiple variables. The results, however, highlight the complexity of a bioarchaeological investigation into the effects of a singular factor on past populations when it is difficult to control for, or even accurately identify, confounding factors. Notable differences in respiratory disease between populations and social groups could, however, be identified, with some links to the consumption of tobacco. Although the results must be interpreted cautiously due to large 95% confidence intervals, tobacco consumers from Barton-upon-Humber did present with a two-fold increase in the likelihood of having respiratory inflammation. This same result was not apparent in the population from London.

While maxillary sinusitis was more prevalent within the population of Barton-upon-Humber compared to St James's, the opposite was observed for pulmonary/pleural inflammation. Maxillary sinusitis was also over two-times more likely to affect people from London from poorer socio-economic backgrounds, although a breakdown of prevalence rates by multiple variables (Fig 5) also indicates that non-smoking males of higher socio-economic status were among the most commonly affected by both upper and lower respiratory inflammation. It is apparent that unique factors within the living environment and differences in activities undertaken by different social groups, including tobacco consumption, have a distinct impact on the prevalence of respiratory disease. These factors may differentially affect the prevalence of maxillary sinusitis and pulmonary/pleural inflammation and may also obscure the impacts of tobacco-use.

It is not uncommon to see differences in the prevalence of respiratory disease between rural and urban archaeological populations [66,71–74]. Regionality, industrialisation, and urbanism (or lack thereof) are likely to have had a strong effect on results from the current study. An increase in pulmonary/pleural inflammation in urban environments compared to contemporaneous rural populations has been found in Sudanese and other London populations [71,75]. One of the major

risk factors contributing to frequencies of pulmonary/pleural inflammation in the St James's population is likely to have been poor air quality. Since the introduction of coal in the medieval period, London has always faced issues with air pollution. Increased usage of coal over time only served to reduce air quality in the city, which became famous for its smoke-thickened fogs and soot-stained buildings. This culminated in the London smogs of the late nineteenth century, events which could cause an excess of deaths reaching over a thousand [76]. We now know that the particulate matter produced by the burning of coal and biomass fuels is particularly harmful to the lower respiratory tract, causing increased pulmonary inflammation and tissue damage, poorer immune response to pathogens, and an increased susceptibility to respiratory tract infection [77].

A further factor of note is the presence of tuberculosis in post-Medieval Britain. The *Bills of Mortality* for London between the seventeenth and eighteenth centuries indicate that between 10 and 20% of all yearly deaths in the city were caused by tuberculosis [78]. Its spread and impact on society were such that, by the nineteenth century, its effects could be seen in culture, fiction, and fashions [79]. A significant association has been found in multiple studies between new bone formation on the ribs and a cause of death related to tuberculosis [80–84]. Although not all individuals demonstrating new bone formation on the ribs in the current study may have died from tuberculosis, a large proportion of the lesions observed may have been the result of pulmonary/pleural inflammation induced by the disease. Further, in countries with a high burden of tuberculosis today, 15.2% of deaths as a result of tuberculosis are attributable to tobacco smoking [85]. Considering that smoking prevalence for these countries is roughly estimated to be around 21% of the population, lower than the frequency of smokers identified in the populations within the current study, it is possible that smoking may have been attributable to an even higher percentage of respiratory-related death in the current study. Further, an association between air pollution and tuberculosis has also been found in modern clinical studies [86], making the environment in post-medieval London particularly conducive for the spread of this disease.

Additionally, while polluting of drinking water in London had been ongoing for centuries, in the nineteenth century, London faced a water sanitation crisis brought to a head by the 'Great Stink' of the summer of 1858 [87], accompanied by high rates of water-borne disease. Unwashed hands in unsanitary environments are a known route of lower respiratory tract infection [88,89]. Waste water has also been found to be a significant route of transmission for tuberculosis pathogens [90]. In addition to this, poor sanitation increases the likelihood of diarrheal disease and a number of parasite infections [91], the presence of which can put pressure on host immunity and increase susceptibility to other infectious diseases. Between 1851 and 1860, Westminster, in which the parish of St James's was situated, had one of the highest mortality rates from diarrhoea in London [92]. Further, St James's was at the heart of a cholera outbreak in 1854 from polluted well water that led to a massive peak in cholera-related mortality in the parish [92]. Both exposure to poor outdoor air quality and unsanitary drinking water by the entire London population between 1700 and 1850 may then have been a 'great equaliser' in terms of socio-economic status and risk of developing pulmonary/pleural inflammation, resulting in similar prevalence rates in different SES groups. Further, such equitable exposure may have also aided in masking the impacts of tobacco consumption on frequencies of respiratory inflammation. In a study of the impacts of tobacco on respiratory health in the Netherlands, a five-times greater odds for maxillary sinusitis was observed in tobacco consumers in the harbour town of Vlissingen, while differences were not observed according to smoking status in the industrialised population of Arnhem [35], where industry primarily involved small-scale production of building materials, shoemaking, beer crafting, and tobacco [93].

In London, air quality and poor sanitation may have generally exposed people to pathogens and particulates in the air below the size of 10 µm (smaller than a grain of pollen), which can penetrate into the lungs. Nonetheless, the upper-classes may have been able to protect themselves from risk factors related to sinusitis, perhaps by installing better fireplaces and affording cleaner burning fuels [76], which may have produced lower levels of larger particulates from incomplete combustion, and from having warmer (i.e., mould-free), well-ventilated homes. Upper- and upper-middle classes may have also avoided exposure to particulates by working in offices or by being able to afford trips to the

countryside for 'fresh air' [76]. Binary logistic regression demonstrated that the lower SES group was almost twice as likely to have maxillary sinusitis than the higher SES group. The prevalence of maxillary sinusitis in the low-middle SES group is more comparable to that found at post-medieval Barton-upon-Humber. These results hint at the possibility that, while higher SES may not have been as effective at buffering against the development of pulmonary/pleural inflammation, factors related to being upper-class potentially provided protection from the development of maxillary sinusitis. Similar differences were observed between the prevalence of maxillary sinusitis in post-medieval London individuals from St Bride's Crypt (higher status) and St Bride's Lower Burial Ground (lower status) [94].

The data also hints at some intersectional impacts. Women from London of lower SES did present with higher frequencies of respiratory disease compared to their higher status counterparts. Non-smoking women of lower status, in particular, appeared to be especially susceptible to pulmonary/pleural inflammation. One possible explanation for this is that the lack of evidence for tobacco consumption in poorer women may actually act as a signifier of those with extremely limited access to resources and commodities, such that the risks from this, including poor nutrition, outweighed the risks from tobacco consumption itself. It is even more difficult, however, to explain why non-smoking men from higher socio-economic backgrounds seem to be disproportionately affected by both upper and lower respiratory inflammation. These results demonstrate the difficulties in interpretation considering the number of confounding factors that contribute towards the development of respiratory diseases.

We must also consider that, although maxillary sinusitis is not necessarily life threatening, many lower respiratory tract infections do, however, lead to mortality. In the 1920s/30s, the majority of deaths for untreated pneumococcal pneumonia occurred within six to ten days [95]. Additionally, today, the risk of death by pneumococcal pneumonia within 30 days in active smokers is five-fold compared to non-smokers [96]. Therefore, in those whose pulmonary health is already at risk (i.e., those with increased frailty), such as tobacco smokers, an acute pulmonary infection may potentially lead to death before new bone can form on the surfaces of the ribs [97]. This may be one of the explanations as to why women from the lower SES group (likely to be one of the most disadvantaged groups during this period) who did not consume tobacco present with higher rates of pulmonary inflammation than their smoking counterparts; because lower-class smoking women did not survive respiratory insults long enough to form evidence of inflammation. Tobacco consumption in combination with limited access of poor women to adequate income, nutrition, and healthcare, the pressures of pregnancy and motherhood, and the numerous risk factors affecting the overall poor, including exposure to pathogens such as tuberculosis and cholera, [98] was likely to have increased frailty. Heterogenous frailty may also apply to the greater prevalence of pulmonary disease in higher status tobacco-consuming men, compared to their non-smoking counterparts, although exposure to risk from other factors was likely to have differed markedly to those borne by lower status women.

Barton-upon-Humber is described as a small market town and population density has always been low. Even during the population 'boom' in the town in the early-nineteenth century, the total population in 1851 only reached 3,866 [43]. While there was a rise in social stratification within the town during the late-eighteenth to nineteenth centuries, with an increase in professional occupations and the erection of more spacious housing [46], this was not as extreme as that occurring in London, ensuring quality of life and access to fresh air may not have been as dependant on socio-economic status. When considering the poor air and water quality in London, it is unsurprising then that pulmonary/pleural inflammation rates are significantly higher within the St James's group, compared to individuals from Barton-upon-Humber. It is also evident that a large proportion of those with pulmonary/pleural inflammation from Barton-upon-Humber consumed tobacco, while consuming and non-consuming groups from St James's were affected relatively equally. All these groups are the most likely to have been directly exposed to poor air quality from either tobacco smoke or other sources, while non-consumers from Barton-upon-Humber (unless breathing in second-hand smoke) may have enjoyed a lower exposure to air pollution.

It is, however, difficult to understand why individuals from Barton-upon-Humber demonstrate a higher prevalence of maxillary sinusitis than the St James's population (although note, this result was no longer significant after *p*-value correction for multiple tests). Evidently, the same risk factors may affect the development of upper and lower respiratory

tract inflammation differently [71]. Agricultural settings have been noted to have a detrimental effect on respiratory health, as they include exposure to particulate matter, toxins, and pathogens. Inflammatory agents can include organic and inorganic matter produced when ploughing, threshing, or winnowing; fungal spores and pollens; and allergens from animal dander (as well as the modern-day practice of pesticide use) [19,99,100]. However, clinical studies have also found that growing up on a farm can actually improve your resistance to upper respiratory tract problems [101]. In rural Britain, mould may have been a particularly prevalent risk factor in rural domestic settings during the wet and cold winters. Women smokers from Barton-upon-Humber demonstrate the highest prevalence rates of maxillary sinusitis, which may be as a result of the combined increased risk from tobacco smoke and exposure to particulates within the household, such as mould, dander, dust, and fire smoke, that women who traditionally spent more time inside the house may have been exposed to. Conversely, male smokers from Barton-upon-Humber presented with the highest rates of pulmonary inflammation from this cemetery. A number of post-medieval trades may have exposed men to particulate matter particularly irritating to the lower respiratory tract, and again, in combination with tobacco consumption, may have greatly increased the risk for pulmonary inflammation. For example, the 1851 census recorded men in the town as builders, rope and sail-cloth makers, quarriers and stone-workers, tanners and leather workers, and brick and tile-makers [43]. Women in 1851 worked as crafters in clothes making and, in 1861, as spinners and weavers in the rope factory. All of these occupations produce either organic or inorganic occupational dusts that can irritate and inflame the respiratory tract. Such dusts from crafts have been proven to be detrimental to respiratory function and cause susceptibility to the development of airway inflammation [102–106].

Additionally, a rise in incidences of respiratory disease between the medieval period and the post-medieval period at Barton-upon-Humber certainly indicates that risk factors increased over time, one of them being the introduction of tobacco consumption. The greatest difference between these groups for pulmonary inflammation was observed between the control medieval group and the post-medieval tobacco consumers. It is not known to what extent post-medieval non-consumers may have been affected by second-hand smoke, but this may also have contributed towards the increase between this group and the medieval control group. Other factors affecting the increase is respiratory disease over time include the swap to brick and tile housing [107], which may have reduced ventilation from indoor fires and tobacco smoke, and the increase in population density. Although not large, a rise in population puts strain on existing clean water resources and sanitation measures. These factors, as well as changes to farming, craft, and trade activities, may have worked in tandem with tobacco consumption to increase risks of developing respiratory disease over time.

## Limitations and future directions

There are a number of additional factors that we have been unable to completely control for within the current study. Major limitations include reduction in sample sizes when breaking groups down by multiple factors, difficulty in determining the medium by which people were exposed to tobacco and to what extent they were exposed, the complexity in determining the aetiology of pathological bone changes, and issues surrounding heterogeneity of frailty within a population.

**Sample size.** In order to investigate the variable impacts of tobacco consumption from an intersectional perspective, individuals must be broken down into groups based on a number of different aspects of their social and biological identity. It is, however, hard to tease out the differences among groups according to tobacco consumption status, estimated age, skeletal sex, and socio-economic status due to the massive reduction in sample size numbers produced as a result of this division. This limitation affected the ability for the current study to assess in-depth the impact of tobacco consumption in conjunction with other factors, particularly in inhibiting the application of hierarchical loglinear analysis. This was particularly notable in the groups from Barton-upon-Humber, which, when broken down by estimated age, skeletal sex, and tobacco consumption status, often presented with groups of less than ten individuals. Mature adults within the St James's group also presented with the same problem. We are, thus, only able to present here potential differences between groups without the application of statistical testing. However, as described above, these results point

to interesting avenues for potential future research that can further explore the impacts of multiple aspects of identity on susceptibility to disease and lived experience.

**Determining tobacco exposure.** Osteoarchaeological approaches for identifying tobacco consumption only provide evidence for direct smoking practices. While we have categorised individuals with no evidence for tobacco consumption as 'non-consumers', we cannot determine whether they were likely to have been unexposed, infrequent consumers, passively exposed to tobacco smoke, or had partaken in snuff (powdered tobacco inhaled via the nose). All we can assume is that these individuals were not habitual smokers to the extent observed in individuals with pipe-notches or lingual staining. Our 'non-consumer' group, therefore, is likely to represent a continuum of exposure that we cannot quantify. Passive smoke exposure (also known as environmental or second-hand smoke exposure) has been linked to decreased respiratory function and increased risk of respiratory symptoms [108–110]. Further, snuff is associated with impaired mucociliary clearance in the nose and chronic rhinosinusitis [111,112], impaired pulmonary function [113] and with increased risk of developing tuberculosis and chronic bronchitis [114]. While these forms of exposure do not always have as high a risk of disease as frequent smoking [115–118], the possible inclusion of individuals with some level of exposure to tobacco within the non-consumer group is likely to have complicated our results. Further, it is also yet unknown whether tobacco inhalation via second-hand smoke, other forms of consumptions, such as snuff, or infrequent consumption creates a metabolic signature similar to that of habitual smokers and is detected in our current biomolecular samples. Future archaeometabolomic research in this area, incorporating modern samples from individuals with known tobacco consumption histories will aid in clarifying this picture.

In London, where people may have spent more time indoors and, in poorer groups, within close quarters, exposure to second-hand smoke may have been a large contributor to respiratory inflammation. This may be one of the reasons why we see less distinction in the disease profiles of smokers and non-smokers between our St James's groups. Further, in English upper-class circles, it was considered impolite for men to smoke around women, resulting in the invention of the 'smoking room' [9], potentially resulting in differential exposure between genders in this SES group. This does not, however, appear to be reflected in the results of the current study, in which high status female non-consumers do not present with notably lower prevalence rates than either high status male consumers or non-consumers.

**The complexity of aetiology.** Although the changes within the maxillary sinuses recorded in the current study are considered to represent sinus inflammation (i.e., sinusitis), the exact aetiology of this inflammation is not easy to discern and may not always be linked directly to environmental exposure. Dental disease, particularly periapical lesion, is a possible cause of maxillary inflammation, known as odontogenic sinusitis [119]. It is, however, impossible to accurately determine whether dental disease is the cause of sinus inflammation in human skeletal remains [120,121]. Further, tobacco consumption also increases the risk of developing oro-dental diseases [122], which in turn may have led to elevated rates of maxillary sinusitis in certain tobacco consuming groups. It was not possible in the constraints of the current study to disentangle the interplay between tobacco consumption, oro-dental disease, and maxillary sinusitis. It should be considered, however, that differences in oro-dental disease among groups, independent of tobacco consumption status, may have had an effect on variability in maxillary sinusitis frequencies. This will be further explored in a forthcoming study.

Further, in the current study, while we have considered new bone formation on the visceral surfaces of the ribs to represent pleural inflammation likely as a result of respiratory disease, it is vital to note that not all instances of new bone formation in this region of the skeleton may be linked to respiratory ailments. Davies-Barrett et al. [71] provide a table of the range of potential conditions that can contribute to pleural disruptions that are likely to cause new bone formation on the ribs, including non-pulmonary aetiologies. Studies of identified human remains collections, with known cause of death, have demonstrated a significant association between new bone formation on the ribs and tuberculosis [80–84]. These same studies have observed a prevalence of 7.4–36.7% in individuals who died from a pulmonary disease other than tuberculosis, but have also observed new bone on the ribs of 2.4–25.0% of individuals who died from a non-pulmonary

disease (including heart disease and cancer). In the pre-antibiotic era, it is likely that a very large portion of individuals in the current study with new bone formation on the ribs reflect pulmonary/pleural inflammation as a result of infectious respiratory disease. For example, at the beginning of the twentieth century, prior to the introduction of antibiotics, the leading causes of death in Britain and the United States were tuberculosis and pneumonia [123,124]. Further, the *London Bills of Mortality*, which provide records of the causes of death within the city between the seventeenth and eighteenth centuries, indicate that at least between 10 and 20% of all yearly deaths in the city were as a result of tuberculosis [78].

There is the potential, however, that not all new bone formation on the ribs observed in the current study may have been caused specifically by respiratory disease, which may complicate the results observed. This also opens up further questions about the role of tobacco consumption in the development of other diseases in the past, such as heart disease, cancer, and systemic inflammation, which have been associated with individuals with new bone formation of the ribs [80,82,84]. The relationship between tobacco consumption and inflammation during the post-medieval period in these same populations will be explored further in a forthcoming study.

**Heterogeneity in frailty.** First highlighted by Wood et al. [97], the osteological paradox raises the issue of heterogeneity in frailty, i.e., that individuals with greater frailty may die from a mortality-causing disease before the production of skeletal lesions, while those with reduced frailty may survive long enough to develop lesions, inhibiting our ability to identify those most at risk of a disease. This is of particular issue when trying to examine groups using an intersectional perspective, in which it might be expected that frailty may be higher in a group exposed to greater marginalisations than in a group exposed to less. One of the major limitations of any bioarchaeological research related to pulmonary/pleural inflammation is the potential for an individual with a highly compromised immune response to die from a pulmonary disease before new bone formation can form on the ribs. Prior to antibiotics, untreated pneumococcal pneumonia in the 1920s/30s caused death on average within six to ten days [95] and, today, active smokers present a 5-fold risk of death from pneumococcal pneumonia within 30 days compared to non-smokers [96]. While the average duration of untreated HIV-negative tuberculosis before death or resolution is around 3 years [125], tuberculosis can also cause acute pneumonia [126].

What we may be seeing in the lower prevalence of pulmonary/pleural inflammation in high-status male and low-status female tobacco consumers is a reflection of heightened frailty as a result of tobacco consumption in tandem with the myriad of risk-factors for poor health in the urban setting, resulting in death prior to new bone forming. This does not explain, however, why low-status males do not also exhibit this pattern and why high-status male non-smokers demonstrate a similar pattern in the frequency of maxillary sinusitis, which is highly unlikely to cause mortality [127], to that of pulmonary inflammation. Further research into the frailty of the population by investigating mortality rates and prevalence rates of other diseases in these same groups may provide further insight.

## Conclusions

Despite King James I's cries against "*a custome lothsome"*, tobacco-use became an intimate part of post-medieval English life, consumed by all facets of society. Current clinical knowledge outlines the undeniable fact that tobacco consumption has innumerable impacts on the body and its healthy functioning. This would have been no less so the case in the past; its physiological impacts were noted by medical practitioners of the day, albeit largely ignored by the public. However, with few to no restrictions on pollutive elements affecting air quality, water sanitation, and occupational health in the post-medieval period, our results indicate that understanding the respiratory health impacts of tobacco consumption is complicated by the numerous other risk factors for respiratory inflammation present. In London, it is apparent that possible differences in respiratory disease prevalence among tobacco consumers and non-consumers might be masked by the impacts of air pollution and unsanitary drinking water, leading to high overall prevalence rates of pulmonary/pleural inflammation. Socio-economic status does, however, seem to have had some effect on a person's susceptibility to maxillary sinusitis, with those of higher socio-economic status likely able to access resources that buffered against risk factors

for the disease. Further, the cumulative impacts over time of poor air quality, tobacco consumption, and other risk factors appear to have affected Londoners, with an increased likelihood of the presence of maxillary sinusitis in later age groups in this population.

Within the rural population, who likely had lower levels of air pollution, population density, and disease load, and possibly more sanitary surroundings, our results do hint at the negative contribution that tobacco consumption may have had on respiratory health. These results have implications for our interpretations of disease prevalence rates in rural post-medieval communities, for which tobacco consumption has not hitherto been considered in-depth as a contributor towards disease frequency. Investigations into tobacco consumption and disease rates in other rural post-medieval communities are required to see if this pattern is replicated elsewhere. Of interest is the significantly higher rate of maxillary sinusitis in the Barton-upon-Humber group, when compared to St James's, and the significant increase in pulmonary inflammation between the medieval and post-medieval periods within the population. It is evident that risk-factors for respiratory disease were prevalent within the later population, one such risk factor potentially being the high percentage of tobacco consumers in the town during the post-medieval period. The agricultural lifestyle and a number of craft-work industries in the town may have also contributed to the inhalation of particulates of a size and composition particularly irritating to the respiratory tract.

There are a number of limitations that have inhibited this research from fully exploring the interrelationship between tobacco consumption and intersectional aspects of identity, and its effects on the expression of respiratory disease profiles. These limitations include the potential impacts of heterogenous frailty on the development of evidence for pulmonary/pleural inflammation, decreased sample sizes when breaking down groups in multiple categories, and difficulties in determining an individual's exposure to tobacco. However, our results hint at potential intersectional impacts on respiratory inflammation, related to consumption status, skeletal sex, socio-economic status, and regionality, that present fertile ground for future studies on this topic.

## Supporting information

**S1 File. Raw data used in the study.**
(CSV)

## Acknowledgments

We would like to gratefully acknowledge the input and support of Diego Badillo-Sanchez and Maria Serrano Ruber for contributions to this research project, and Jo Appleby and the University of Leicester Bioarchaeology Research Cluster for providing feedback on a previous draft of this manuscript. We would like to thank Simon Mays, Kevin Booth, Andrea Bradley, Hayley James, Michael Henderson, and Louise Fowler for facilitating access to the skeletal individuals included in this study. We would also like to thank Raguwinder (Bindy) Sahota and Karl Storz SE & Co. KG for access to the endoscope used in this study. We would also like to thank our five reviewers, whose valuable comments contributed towards improving the quality of our manuscript.

## Author contributions

**Conceptualization:** Anna M. Davies-Barrett, Sarah A. Inskip.

**Data curation:** Anna M. Davies-Barrett.

**Formal analysis:** Anna M. Davies-Barrett, Maia Casna.

**Funding acquisition:** Sarah A. Inskip.

**Investigation:** Anna M. Davies-Barrett.

**Methodology:** Anna M. Davies-Barrett.

**Project administration:** Anna M. Davies-Barrett, Sarah A. Inskip.

**Visualization:** Anna M. Davies-Barrett.

**Writing – original draft:** Anna M. Davies-Barrett.

**Writing – review & editing:** Anna M. Davies-Barrett, Maia Casna, Sarah A. Inskip.

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
