## [Decision Letter · Decision Letter 0]

22 Jan 2025

PONE-D-24-53580“A custome lothsome”: Investigating the association between tobacco consumption and respiratory inflammation in two post-medieval English populations (c. CE 1500–1855)PLOS ONE

Dear Dr. Davies-Barrett,

Thank you for submitting your manuscript to PLOS ONE. After careful consideration, we feel that it has merit but does not fully meet PLOS ONE’s publication criteria as it currently stands. Therefore, we invite you to submit a revised version of the manuscript that addresses the points raised during the review process.

**ACADEMIC EDITOR:**

Dear Dr. Davies-Barrett,

We appreciate you submitting your manuscript to PLOS ONE and thank you for giving us the opportunity to consider your work.<o:p></o:p>

I have completed my evaluation of your manuscript, which has been reviewed by five highly qualified reviewers all of whom agree it is worth to be published in PLOS ONE. Nevertheless, they have suggested changes that will help to improve the paper.<o:p></o:p>

Therefore, I invite you to resubmit your manuscript after addressing the reviewers’ comments below. When revising your manuscript, please consider all issues mentioned in the reviewers' comments carefully: please, outline every change made in response to their comments and provide suitable rebuttals for any comments not addressed. Please, note that your revised submission may need to be re-reviewed.<o:p></o:p>

PLOS ONE values your contribution and I look forward to receiving your revised manuscript.<o:p></o:p>

Yours sincerely,<o:p></o:p>

Dr. Olga Spekker

We look forward to receiving your revised manuscript.

Kind regards,

Olga Spekker, Ph.D.

Academic Editor

PLOS ONE

Journal requirements: When submitting your revision, we need you to address these additional requirements. 1. Please ensure that your manuscript meets PLOS ONE's style requirements, including those for file naming. The PLOS ONE style templates can be found at https://journals.plos.org/plosone/s/file?id=wjVg/PLOSOne_formatting_sample_main_body.pdf and https://journals.plos.org/plosone/s/file?id=ba62/PLOSOne_formatting_sample_title_authors_affiliations.pdf. 2. Please amend either the title on the online submission form (via Edit Submission) or the title in the manuscript so that they are identical. 3. In your manuscript, please provide additional information regarding the specimens used in your study. Ensure that you have reported human remain specimen numbers and complete repository information, including museum name and geographic location.  If permits were required, please ensure that you have provided details for all permits that were obtained, including the full name of the issuing authority, and add the following statement: 'All necessary permits were obtained for the described study, which complied with all relevant regulations.' If no permits were required, please include the following statement: 'No permits were required for the described study, which complied with all relevant regulations.' For more information on PLOS ONE's requirements for paleontology and archeology research, see https://journals.plos.org/plosone/s/submission-guidelines#loc-paleontology-and-archaeology-research. 4. We note that the grant information you provided in the ‘Funding Information’ and ‘Financial Disclosure’ sections do not match.  When you resubmit, please ensure that you provide the correct grant numbers for the awards you received for your study in the ‘Funding Information’ section. 5. Thank you for stating the following financial disclosure:  [This research was mainly funded by a UKRI-FLF research grant (grant no.: MR/T022302/1), awarded to SI. Additional funding came from a Dutch Research Council (NWO) grant (grant no.: PGW.21.008), awarded to MC.].  Please state what role the funders took in the study.  If the funders had no role, please state: ""The funders had no role in study design, data collection and analysis, decision to publish, or preparation of the manuscript."" If this statement is not correct you must amend it as needed. Please include this amended Role of Funder statement in your cover letter; we will change the online submission form on your behalf. 6. Your ethics statement should only appear in the Methods section of your manuscript. If your ethics statement is written in any section besides the Methods, please move it to the Methods section and delete it from any other section. Please ensure that your ethics statement is included in your manuscript, as the ethics statement entered into the online submission form will not be published alongside your manuscript.  7. We are unable to open your Supporting Information file [Supplemental File S1 - Raw data.sav]. Please kindly revise as necessary and re-upload.

Reviewers' comments:

Reviewer's Responses to Questions

**Comments to the Author**

1. Is the manuscript technically sound, and do the data support the conclusions?

Reviewer #1: Yes

Reviewer #2: Yes

Reviewer #3: Partly

Reviewer #4: Partly

Reviewer #5: Yes

2. Has the statistical analysis been performed appropriately and rigorously? 

Reviewer #1: Yes

Reviewer #2: Yes

Reviewer #3: Yes

Reviewer #4: Yes

Reviewer #5: Yes

3. Have the authors made all data underlying the findings in their manuscript fully available?

Reviewer #1: Yes

Reviewer #2: Yes

Reviewer #3: Yes

Reviewer #4: Yes

Reviewer #5: Yes

4. Is the manuscript presented in an intelligible fashion and written in standard English?

Reviewer #1: Yes

Reviewer #2: Yes

Reviewer #3: Yes

Reviewer #4: Yes

Reviewer #5: Yes

5. Review Comments to the Author

Reviewer #1: Dear authors,

this is a very well-written manuscript and well-planned research. I appreciate the opportunity given to read your manuscript. I have only minor observations that the authors should consider, they mostly express a concern with the need to avoid over simplification views of what can be inferred based on the analysis of human remains

Por example, in line 52 “Today, smoking is synonymous with concepts of blackened, diseased lungs and poorly, wheezing individuals.” - this is a very general description, please provide more detail, or be more specific. Throughout the manuscript, often, some arguments offer a very generalist view of the facts - this should be avoided.

in Lines 71 - 83: although adequate, this section ignores the relevance of differential diagnosis when exploring bone changes. The authors provide an argument that focuses on the relation between smoke and disease-related bone changes, failing to address the complexity of disease-related bone changes etiology.

As the authors must certainly know the presence of high levels of prevalence and/or high correlation is not necessarily a synonymy of causation. This needs to be brought to the Introduction and discussion.

Also, please state if you have permission to reproduce figure 1, even it being an adaptation, this needs to be clear.

Reviewer #2: The manuscript presents new data and important new insights into respiratory health and the impact of tobacco consumption in the past. The statistical tests applied are appropriate for the sample based on the kind of data collected and the size of the cohorts, and the tables and figures are clear and well presented. There are a few points which the authors may wish to elaborate; a sentence or two giving a working definition of intersectionality as it is understood in biological anthropology would be very useful at the outset. Following this, for example in lines 380ff, it would be interesting to explore these data in light of the framework for intersectionality you have presented; only a line or two about what these confounders might be would suffice (which you have done with discussion of other cohorts).

The following is a line-by-line review of some typos and stylistic edits that could be applied.

ll. 16 : assessed

ll. 21: over two times - could become : more than twice as likely

ll. 28: delete comma after of

ll. 59: remove commas

ll. 66: replace : to the risks, with, of the risks

ll. 67: delete against (this is already implied in the meaning of the word decry)

ll. 122ff: ensure that there is a space after the date and before CE

ll. 176: suggest starting this line with, A discussion of the pathophysiological processes...

ll. 203-204 : italicise p in p-value

ll. 206: introduce the abbreviation of odds ratio (OR)

ll. 371: The data also hints at...(to avoid use of the passive voice here)

ll. 184 and 510: The full trading name of Karl Storz should be represented here: Karl Storz SE & Co. KG, and perhaps city and country if required by journal submission guidelines.

Reviewer #3: The present work is an interesting attempt to analyze the impact of smoking on human health using osteoarchaeological and metabolite analyses of English citizens during the post-medieval period. Overall, the paper is well-prepared; however, in my opinion, two issues are not fully addressed and require further consideration:

1. Sample size: The study relies on a relatively small sample size, especially when the examined groups are further subdivided based on multiple variables (tobacco consumption, age, sex, and socioeconomic status). This issue is particularly pronounced for the Barton-upon-Humber site, where subdividing the sample results in very small groups (N<10; see Tables 1 and 2). As a result, the ability to draw reliable conclusions from the data is limited. Did the authors consider including additional samples, particularly from the pre-tobacco period, to increase the sample size?

2. Controlling for confounding factors: The authors’ assumption that maxillary sinusitis and pulmonary/pleural inflammation can be directly linked to tobacco use, despite the multifactorial etiology of these changes, is questionable. While the authors acknowledge this issue in the Discussion section and suggest it as a reason for the ambiguous results, it remains unclear how they intended to control for confounding factors (if this is even possible, which I sincerely doubt) and demonstrate that the observed respiratory diseases were specifically caused by tobacco smoking rather than other factors.

Other comments:

• The Material section lacks information on the percentage of individuals excluded from the study due to the lack of suitable skeletal elements or the poor state of preservation of the human skeletal remains. Please include this information.

• The Methods section lacks a brief description of how the molecular analysis was performed. Simply stating that "metabolomic identification of small molecules (metabolites)" was used is insufficient, especially since this method is not commonly employed in osteoarchaeological research. While citations for the methods are provided in the Introduction, they are absent from the Methods section, except for those concerning the signatures used. Additionally, there is no information regarding the research laboratory protocol followed during the analysis. Were all steps carried out as described in the paper by Badillo-Sanchez et al. (2024)? For what percentage of all individuals was metabolite analysis performed? Were the analyses successful for all these individuals?

• It is important to emphasize that exposure to tobacco smoke, and not just active smoking, induces changes in the sinuses and ribs. How can second-hand smokers be distinguished in osteoarchaeological studies? If such identification is not possible, how can it be ensured that the authors have correctly classified individuals exposed to tobacco smoke versus those who were not? This issue is particularly significant, especially since "it is yet unknown whether tobacco inhalation via second-hand smoke (…) creates a metabolic signature similar to direct smokers". Please clarify.

• How were data on individuals classified as "possible" and "probable" women or men interpreted? Were they included in the results as women/men, even though the assessment was uncertain? If so, could this have contributed to the ambiguous results? Additionally, which numbers in the sex groups in the supplement correspond to these categories? This has not been clearly explained in the text.

• The authors provide an insightful overview of the importance of tobacco in post-medieval England. However, the manuscript does not address whether regional variations in tobacco use existed during this period. Is this information available?

Abstract: The word "assed" should be corrected to "assessed."

Discussion: The sentence "Some intersectional impacts are also hinted at." lacks clarity and specificity.

Figure 1: Map lacks a scale, making it difficult to accurately assess the distance between sites.

Figure 4: Does the y-axis represent the frequency or indicate the number of individuals within each category?

Reviewer #4: The topic of the paper manuscript is interesting and as stated the effect of tobacco on health in past populations is not well understood, so the paper rightfully offers to lay a steppingstone towards understanding this. However, I find that the paper too comprehensive not sticking to the major focus (see comments) and I think will benefit greatly from restructuring and taking out parts of analysis. I have some concerns and general and more specific recommendations for changes:

General comments

First, the data used is not ideal for the analysis made and conclusions that is sought drawn from them. It is a problem that no medieval ‘baseline’ sample is included for the urban site and the baseline sample from the rural site is very small (especially when grouped into sex and age groups). Without also including a urban 'baseline' I would exclude the medieval 'baseline', - it does not really bring much information and is not used actively in overall conclusions. Furthermore, the post-medieval rural population covers the time period 1500 – 1855 which is very wide and thus not very comparable to the chronologically narrow urban post-medieval population dated within 1789-1853. If possible to group the chronology of the rural site into early and late post-medieval period it would be very beneficial to the comparative analysis of the two sites. Furthermore, in general the samples used are as pointed out by the authors very small when divided into sex, age groups, non-smoker/smoker, social status, chronological groups and urban/rural and therefore not much can actually be concluded.

Regarding the methods for assessing smoking status. Have you considered whether levelling how much you smoke would be important for the results of the study and also could the degree of smoking be divided into levels using the applied methods for distinguishing smokers from non-smokers. A mention of this either in regards to methods or in the discussion would be an interesting addition to the perspectives of the research topic.

Also, it is stated (lines 12-14) that; the paper will explore the potential respiratory health implications of the rapid incorporation of tobacco-use into the everyday lives of English citizens during the post-medieval period but a study of the general prevalence of the respiratory diseases without looking at smoking status is also included. This results in a presentation of results and discussion that is very confusing to read and leaving the reader unsure about what exactly is being discussed and whether results, discussion or conclusions are minded on non-consumers vs. consumers or just generally on SES groups, dating groups or rural/urban. This is in my opinion something not directly related to the purpose of the paper and the paper would benefit from leaving the prevalence of the respiratory diseases out when not studied in relation to non-smokers and smokers. This would also contribute to solving some of the problems with small samples size because in the present version the small groups and many different potential confounding factors sought explored makes the conclusions very weak, and in many instances not reliable as no statistical test results to back them up. I think the authors in general take conclusions too far without stats to support them.

I think much more emphasis is needed on discussing tuberculosis as a major confounding factor for the respiratory disease indicators especially indicators of pulmonary infection. This disease is known as highly prevalent especially from late 18th century and in the analysis that shows no differences among smokers and non-smokers an explanations could be that many where infected with tuberculosis with active disease affecting the respiratory system especially the longs resulting in periosteal reaction on visceral surface of ribs. Tuberculosis is only mentioned shortly in line 338 in connection to transmission from wastewater.

And also too little discussion and emphasis on interpretation of results in the light of the osteological paradox in regards to the prevalence of bony indicators of both maxillar sinus infection and pulmonary infection. Both conditions (or implications from co-infection) could very likely cause death before bone involvement which will have a major impact on the possibilities to draw the conclusions made. This is mentioned briefly in line 391 in relation to lower-class smoking women that did not survive respiratory insults long enough to form evidence of inflammation, but this deserves a section on its own.

Tables 1 and 2: No statistical tests of the differences in prevalence rates?

Figure 5: Confusing, - gives only prevalence rates and no counts or statistical test results therefore difficult to assess the significance of the distributions shown. Need to turn to large tables 1 and 2 for counts which is confusion for the reader. I think the figure is misleading because the lack of significance of the differences of the distributions cannot be seen in the figure.

Specific comments

Line 14: Adult skeletons

Line 16: assessed not assed

Lines 68, 69, 70: Referencing should be listed in end notes as for the remaining manuscript

Lines 72 – 75: will the absence of pipe-notches and staining on dentition be evidence of non-smoking?

Lines 89-93: Very long and a bit unclear - consider rewriting and splitting up in two sentences.

Lines 99-100: If not tested against smoking status how does this give insights into the effect of tobacco smoking on the health in the past?

Line 100: (see 37) should be changed to (37).

Line 115: what date/definition of post-medieval period are you using?

Line 125-126: 62 skeletons (note the sheet with raw data provided has 60 medieval skeletons listed) that later are split into males and females, is a very small sample for providing a ‘baseline’ for pre-tobacco respiratory disease prevalence and a problem that this ‘baseline’ sample only is provided for the rural population because later rural and urban differences are discussed.

Line 156-157: (see 4) changed to (4) or identified according to the methods for registration presented by Davies-Barret and Inskip (4). How certain will a non-smoking status be without any of these indicators?

Line 157ff: it is mentioned that when not possible to identify tobacco consumption status from osteoarchaeological methods biomolecular methods were applied – why were biomolecular methods not applied to all?? When data about smoking status not collected in the same manner what implications for final results?? Please write in main text how many with tobacco consumption status assessed from dentition and how many from biomolecular analysis.

Lines 166ff: I know these are standard methods and widely used, but it is unfortunate that old and not very accurate methods for aging are used when newer methods are available – ex. the latest version of transition analysis (https://www.statsmachine.net/software/TA3/).

Line 169: Also not very ideal way of grouping (but know it is used as a standard) here the age group 50+ is rather useless when some individuals in the samples likely where much older (70, 80 or 90) when they died.

Line 177: miss be between can and found.

Lines 178-180: and what were criteria for scoring? Whenever pathology present or certain amount?

Lines 230: Statistical significance? And what table to look for results?

Lines 263-264: compared to non-consumers?

Lines 263ff: Results presented have no statistical test results and conclusions cannot be made from the mentioned rates. Also, rates presented are a mix of individuals where no information about tobacco consumption status and with information. This is very confusing and as mentioned before I find that parts where no information about smoking status is present not really are relevant to include because they are beyond the purpose of the paper, - the results and later discussion should be kept within the purpose and aims of the paper.

Lines 272ff: Again results for groups without information about smoking status is included and do not add to insights into the relation of tobacco consumption and respiratory diseases.

Line 274: 2.8 should be changed to 2.7 (according to table 4)

Lines 314-315: non-smoking males of higher socio-economic status most affected by respiratory inflammation – a very likely explanation is that they survived longer than their smoker counterparts and thus lived long enough for bone changes related to the inflammatory diseases to develop and therefore not necessarily had a higher prevalence of disease.

Lines 315-316: Here I believe a very important confounding factor is tuberculosis infection and this deserved a mentioning and also elaboration.

Lines 321: Unfortunately the problem with the sampling makes it very difficult to test either of these potential effects in detail.

Line 323ff: Again tuberculosis needs to be mentioned.

Sections: line 355ff, line 371ff, line 382ff and line 393ff, and line 407ff: have many parts in which discussion of respiratory disease in general is made but without the perspectives of tobacco consumption. In my opinion when not really linked to results of the paper not involving tobacco consumption this is not relevant for the current paper.

Lines 432-434: Or the fact that general health improves and life expectancy increase and people live longer with respiratory disease and develop more bone changes.

Lines 456-460: therefore suggest leaving out some of these potential factors to get more clear picture of the impact of tobacco consumption.

Line 488-489: Or they lived long enough to develop bone changes.

Reviewer #5: The submitted paper presents an application of biochemical data to palaeopathological explorations of respiratory health with consideration of whether tobacco consumption affects the prevalence of osteological evidence for respiratory disease. The results are interesting and highlight the complexities of identifying and adequately accounting for increasing environmental risk factors for disease that occurred with intensive industrialisation and urbanisation and how these can mask other factors that may present clearer data in other settings.

A number of comments are presented below which I think would improve the presented work in its clarity for readers.

Abstract

1- Page 1, Line 16: Typo identified – original text “Individuals were assed”. Assume this is meant to say, “Individuals were assessed”.

Main Text

2- Page 2, Line 44: Original text “While tobacco-use occurred frequently across all strata of society, the way people consumed became”. Suggested change “the way people consumed tobacco”. The sentence will read easier with it being explicitly being stated what people were consuming.

3- Page 3, Lines 54 – 62: Original text: “It induces respiratory inflammation, damages the tissues of the respiratory tract ….. oral diseases (23-26), to name but a few conditions associated with consumption of the drug.” This section has a repetitive feel to it due to the first two sentences both stating that the consumption of tobacco will increase infections/infectious diseases.

The opening sentence of this section of text could be improved by either a) removing the reference to infection (“promotes infection”) or b) moving this towards the end of the sentence and mentioning non-infectious disease too (“tract, increasing susceptibility to a range of infectious and non-infectious respiratory diseases”.

The start of the second sentence should also be reworked to remove “increases the risk of” portion as this will help reduce the sense of repetitiveness that currently exists across the first three sentences (lots of use of “increase”/”increased”).

4- Page 4, Line 76: Original Text “Inskip and colleagues (32). There is inconsistent use of “and colleagues” versus “et al.” (used on line 79) through the paper.

5- Page 6, Figure 2, Caption: The reference for this figure (Davies-Barrett et al., in review) is not listed in the references at the end of the paper.

6- Page 6, Line 139: Original text “‘rate payers’ (those who owned property)”. Please define rate payers in line 134 instead as that is the first use of the term in the paper.

7- Page 9, Lines 209 to 212: Original text “In binary logistic regression, if the upper limit of the 95% confidence interval falls below one”. The use of “one” (in text form) in this sentence and the sentence explaining the lower limit is unclear. Does the text mean that the 95% confidence intervals between groups are compared, or does the text mean that the limit of the falls above/below the numerical value of 1? If it is the latter, “one” should be written as “1” so it is clear that a numerical value is referred to here.

8- Page 9, Line 217 and Figure 4: Original text “However, a lower frequency of mature adults”. Frequency refers to the rate of something in a period of time or sample and so should not be used to refer to a number which is not a rate. Please change the use of “frequency” in this line of text to “number” and to “Number of individuals” on the y-axis of the figure.

9- Page 14, Figure 5: This is a really interesting way of visually representing the collective results from the study. I wonder whether numerical labels on the circles would help with the readability of the figure e.g. if 0°-180° is vertical with 0° at the top of the circle, 0° would have 0%, 90° and 270° would have 50% and 180° (bottom of the circle) would have 100%).

10- Page 20, Line 408: Original text “(although not this result was)”. Assumed typo and “not” is meant to be “note”.

6. PLOS authors have the option to publish the peer review history of their article (what does this mean? ). If published, this will include your full peer review and any attached files.

**Do you want your identity to be public for this peer review?** For information about this choice, including consent withdrawal, please see our Privacy Policy .

Reviewer #1: No

Reviewer #2: No

Reviewer #3: No

Reviewer #4: No

Reviewer #5: **Yes: ** Matthew J. Lee

---

## [Author Response · Author response to Decision Letter 1]

11 Mar 2025

A 'Response to reviewers' file has been uploaded. responses to all comments can be found within this document.

---

## [Decision Letter · Decision Letter 1]

15 Apr 2025

PONE-D-24-53580R1“A custome lothsome”: Investigating the association between tobacco consumption and respiratory inflammation in two post-medieval English populations (c. CE 1500–1855)PLOS ONE

Dear Dr. Davies-Barrett,

Thank you for submitting your manuscript to PLOS ONE. After careful consideration, we feel that it has merit but does not fully meet PLOS ONE’s publication criteria as it currently stands. Therefore, we invite you to submit a revised version of the manuscript that addresses the points raised during the review process.

**ACADEMIC EDITOR:**Dear Dr. Davies-Barrett,

We appreciate you submitting your manuscript to PLOS ONE and thank you for giving us the opportunity to consider your work.

I have completed my evaluation of your manuscript, which has been reviewed by four highly qualified reviewers all of whom agree it is worth to be published in PLOS ONE. Nevertheless, two of the reviewers have suggested some very minor changes that should be considered.

Therefore, I invite you to resubmit your manuscript after addressing the reviewers’ comments below.

PLOS ONE values your contribution and I look forward to receiving your revised manuscript.

Yours sincerely,

Dr. Olga Spekker

We look forward to receiving your revised manuscript.

Kind regards,

Olga Spekker, Ph.D.

Academic Editor

PLOS ONE

Journal Requirements:

Reviewers' comments:

Reviewer's Responses to Questions

**Comments to the Author**

1. If the authors have adequately addressed your comments raised in a previous round of review and you feel that this manuscript is now acceptable for publication, you may indicate that here to bypass the “Comments to the Author” section, enter your conflict of interest statement in the “Confidential to Editor” section, and submit your "Accept" recommendation.

Reviewer #1: All comments have been addressed

Reviewer #3: (No Response)

Reviewer #4: All comments have been addressed

Reviewer #5: All comments have been addressed

2. Is the manuscript technically sound, and do the data support the conclusions?

Reviewer #1: Yes

Reviewer #3: Yes

Reviewer #4: (No Response)

Reviewer #5: Yes

3. Has the statistical analysis been performed appropriately and rigorously? 

Reviewer #1: Yes

Reviewer #3: Yes

Reviewer #4: (No Response)

Reviewer #5: Yes

4. Have the authors made all data underlying the findings in their manuscript fully available?

Reviewer #1: Yes

Reviewer #3: Yes

Reviewer #4: (No Response)

Reviewer #5: Yes

5. Is the manuscript presented in an intelligible fashion and written in standard English?

Reviewer #1: Yes

Reviewer #3: Yes

Reviewer #4: (No Response)

Reviewer #5: Yes

6. Review Comments to the Author

Reviewer #1: Dear authors,

Thank you for taking the time to address all the comments. I have carefully read the reply and the added/expanded text in the manuscript. The comments were addressed with sucess.

Reviewer #3: I appreciate the efforts of the authors in preparing the revised version of the manuscript. The authors have addressed all comments and issues raised during the review process, resulting in significant improvements to the manuscript.

Although the issue of the limited sample size remains unresolved, it has been appropriately acknowledged and discussed in the Limitations section. In addition, the manuscript has been strengthened with thoughtful suggestions for future research.

This study provides a valuable contribution to the ongoing discourse on respiratory diseases and their potential association with tobacco consumption. In my opinion, the work by Davies-Barrett et al. is suitable for consideration for publication in PLOS One.

However, prior to publication, the manuscript should undergo careful language and editorial review. For example:

• Line 450, p. 22: Insert appropriate spacing around 10 µm, which is currently lacking.

• Line 478, p. 23: Amend the punctuation in “… deaths for untreated pneumococcal pneumonia occurred within six to ten days [95], Additionally, …” – a full stop should be placed after the citation: [95]. Additionally,

• Lines 622 (p. 28) and 624 (p. 29): In “7.4 – 36.7%” and “2.4 – 25.0%”, remove the surrounding spaces around the en dash.

• Line 638, p. 29: Remove the outer parentheses in ([80,82,84]) if already enclosed in square brackets, as per the citation style (i.e., should appear as [80,82,84]).

Reviewer #4: I find that the authors have addressed the comments from reviewers adequately, and I look forward to see the published paper. Good luck with your further research within this topic.

Reviewer #5: Thank you to the authors for revising their manuscript and providing detailed responses to the points raised by the reviewers. I am happy with the changes that have been made both regarding the comments I raised, and those my fellow reviewers raised, and I would be happy to see the article published in its current state.

I would suggest one minor edit to the authors, but this is not an impediment to publication.

I feel the repetition of stating that analysis took place in the Human Osteology Lab at Leicester (lines 144/145 and 167/168) could be streamlined into a single line stating that “all analysis occurred in the Human Osteology Lab…..”. Whilst I personally do not think they are required information, I can understand that in the context of stating repository and curatorship information (journal requirement 3) of the human remains, it makes sense to highlight that analysis did not occur in the storage locations. But it would be simpler to state the analytical location once rather than repeating it.

7. PLOS authors have the option to publish the peer review history of their article (what does this mean? ). If published, this will include your full peer review and any attached files.

**Do you want your identity to be public for this peer review?** For information about this choice, including consent withdrawal, please see our Privacy Policy .

Reviewer #1: No

Reviewer #3: No

Reviewer #4: No

Reviewer #5: **Yes: ** Matthew James Lee

---

## [Author Response · Author response to Decision Letter 2]

16 Apr 2025

Please see attached document for our responses to the reviewers.

---

## [Editor Report · Decision Letter 2]

20 Apr 2025

“A custome lothsome”: Investigating the association between tobacco consumption and respiratory inflammation in two post-medieval English populations (c. CE 1500–1855)

PONE-D-24-53580R2

Dear Dr. Davies-Barrett,

We’re pleased to inform you that your manuscript has been judged scientifically suitable for publication and will be formally accepted for publication once it meets all outstanding technical requirements.

Kind regards,

Olga Spekker, Ph.D.

Academic Editor

PLOS ONE
---

## [Editor Report · Acceptance letter]

PONE-D-24-53580R2

PLOS ONE

Dear Dr. Davies-Barrett,

I'm pleased to inform you that your manuscript has been deemed suitable for publication in PLOS ONE. Congratulations! Your manuscript is now being handed over to our production team.

Kind regards,

on behalf of

Dr. Olga Spekker

Academic Editor

PLOS ONE